# HUME: Measuring the Human-Model Performance Gap in Text Embedding Tasks

**Adnan El Assadi**
Carleton University

**Isaac Chung**
Zendesk

**Roman Solomatin**
SaluteDevices, MIRAI

**Niklas Muennighoff**
Stanford University

**Kenneth Enevoldsen**
Aarhus University

## Abstract

Comparing human and model performance offers a valuable perspective for understanding the strengths and limitations of embedding models, highlighting where they succeed and where they fail to capture meaning and nuance. However, such comparisons are rarely made, as human performance on embedding tasks is difficult to measure. To fill this gap, we introduce HUME: **Hum**an **E**valuation Framework for Text Embeddings. While frameworks like MTEB provide broad model evaluation, they lack reliable estimates of human performance, limiting the interpretability of model scores. We measure human performance across 16 MTEB datasets spanning reranking, classification, clustering, and semantic textual similarity across linguistically diverse high- and low-resource languages. Humans achieve an average performance of 77.6% compared to 80.1% for the best embedding model, though with substantial variation: models reach high performance on some datasets while struggling on notably low-resource languages. Our human annotations also reveal multiple dataset issues. We additionally benchmark nine LLMs as annotators on reranking, classification, and STS tasks, finding that they fall short of human performance (76.1% vs. 81.2%) despite offering scalability advantages. We provide human performance baselines, insights into task difficulty patterns, and an extensible evaluation framework that enables a more meaningful interpretation of results and informs the development of both models and benchmarks. Our code, dataset, and leaderboard are publicly available at `https://github.com/embeddings-benchmark/mteb`.

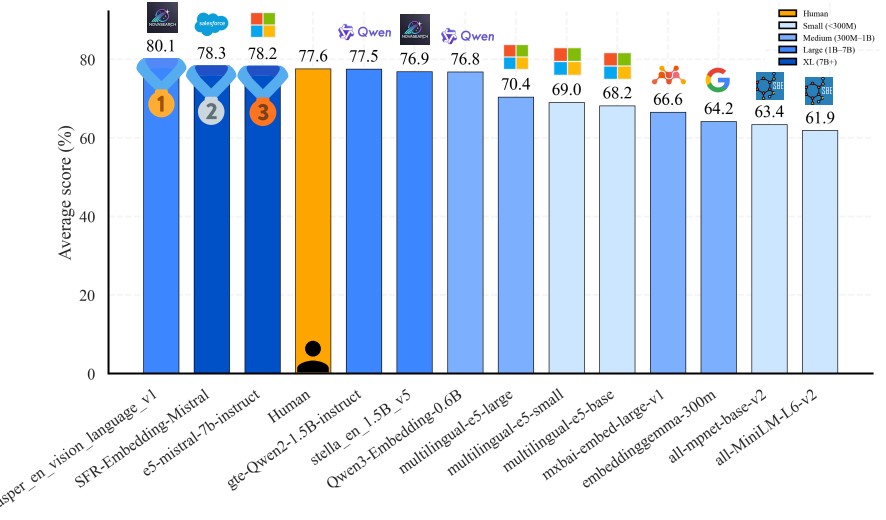

Figure 1: Human performance versus 13 embedding models across 16 tasks. Humans rank 4th (77.6), showing competitive but not dominant performance. Darker shades indicate larger models.

# 1 INTRODUCTION

Embedding models are central to modern NLP systems, powering applications such as search, recommendation, semantic analysis, and information retrieval. Many benchmarks test the performance of embedding models, with the most comprehensive offering a diverse suite of tasks that test their generality and robustness (Muennighoff et al., 2022; Xiao et al., 2025; Enevoldsen et al., 2025). Despite these advances, the interpretation and quality of these scores are often unclear as it is absent of human performance references. Current metrics define performance in terms of theoretical maxima (e.g., MAP = 1.0) that assume perfect consensus on task outcomes. However, many NLP tasks inherently involve ambiguity and disagreement (Plank, 2022), making a model's score difficult to meaningfully interpret without reasonable references. This interpretability gap has serious consequences. When benchmarks reward models for fitting noisy labels, labels where even human annotators disagree, the field risks "blind optimization": expending R&D resources to replicate annotation artifacts rather than achieve semantic progress. For instance, if a model achieves 0.85 MAP in reranking, it is unclear whether this should be considered strong, mediocre, or beyond what annotators typically achieve. This disconnect highlights the need for human-centered evaluation that contextualizes benchmark results. Importantly, human performance should not be treated as an upper bound but as a diagnostic signal: a way to understand where tasks are inherently noisy, where models may surpass typical annotator agreement, and where model behaviour diverges from human judgment.

To address this, we introduce **HUME**: a **Hum**an **E**valuation framework for text embedding tasks. HUME evaluates annotator performance across four task categories: reranking, classification, clustering, and semantic textual similarity (STS), using 16 diverse datasets from the Massive Text Embedding Benchmark (MTEB) (Muennighoff et al., 2022), adapted for human annotation feasibility. Through multi-annotator experiments, we analyze task difficulty, quantify variation across humans, and compare results directly against state-of-the-art embedding models.

Our contributions are threefold: (1) a generalizable framework for human evaluation of embedding tasks, (2) empirical evidence of how humans perform across diverse datasets and task types, and (3) comparative analysis of models and humans that highlights strengths, weaknesses, and ambiguities in both benchmarks and models that yield actionable insights. Together, these contributions establish a foundation for human-aligned evaluation of embedding models and guide future benchmark design.

Beyond human evaluation, we also investigate whether Large Language Models (LLMs) can serve as scalable proxies for human judgment. The promise of LLM-as-annotator approaches is compelling: if LLMs can reliably replicate human judgments, they could enable large-scale benchmark development at lower cost. However, this assumes LLMs capture the same semantic distinctions humans make, rather than exhibiting systematic biases. We evaluate nine state-of-the-art LLMs (GPT-5, GPT-4.1, Gemini, Mistral, and Qwen3 variants) on identical annotation tasks to assess their viability as human proxies and identify task-specific limitations.

# 2 RELATED WORK

## 2.1 TEXT EMBEDDING MODELS AND MTEB

Text embedding models map natural language into dense vectors that capture semantic information. They have progressed from static embeddings (Mikolov et al., 2013; Pennington et al., 2014) to contextual encoders (Devlin et al., 2019; Liu et al., 2019) and more recently to models optimized for embeddings like Sentence-BERT (Reimers & Gurevych, 2019), E5 (Wang et al., 2022), and GTE (Li et al., 2023), powering applications such as search, classification, and clustering.

To evaluate these models across its diverse use-cases, MTEB (Muennighoff et al., 2022; Enevoldsen et al., 2025) has risen as the de facto benchmark framework for embeddings and consolidates evaluation across diverse tasks and datasets. Despite its breadth and community-driven extensions – spanning multilingual, multimodal, and domain-specific variants (Xiao et al., 2025; Kasmaee et al., 2025; Tang & Yang, 2025; Xiao et al., 2024; Ciancone et al., 2024; Enevoldsen et al., 2024; Zinvandi et al., 2025; Wehrli et al., 2024; Poświata et al., 2024; Snegirev et al., 2024), MTEB lacks human performance baselines, making it difficult to contextualize model achievements.

## 2.2 HUMAN EVALUATION IN NLP

Human evaluation is well established in NLP, especially for generative tasks like machine translation (Graham et al., 2013), summarization (Fabbri et al., 2021), and dialogue (Gupta et al., 2019). In contrast, embedding-based tasks have relied almost exclusively on automated metrics, with little attention to human baselines.

In information retrieval, initiatives such as TREC (Voorhees & Tice, 2000) collect human relevance judgments, but these serve as gold standards rather than benchmarks of human performance under model metrics (e.g., nDCG, MRR). Similarly, GLUE, SuperGLUE (Wang et al., 2018; 2019), and MERA (Fenogenova et al., 2024) – a Russian GLUE-like benchmark – report human baselines, but mainly for classification and reasoning tasks. For embeddings, works like STS (Cer et al., 2017) report inter-annotator agreement, yet these are not converted into model-comparable performance scores. This leaves a gap: human performance on embedding benchmarks such as MTEB remains largely unquantified.

## 3 METHODOLOGY

### 3.1 FRAMEWORK DESIGN

Our framework builds on MTEB by establishing reproducible human evaluation protocols that align directly with model evaluation. It consists of task-specific annotation interfaces, principled dataset sampling, a standardized results format, and the use of aligned metrics. This design reveals where evaluation practices introduce ambiguity or inconsistency.

### 3.2 TASKS, DATASETS, AND METRICS

Our selection criteria ensures comprehensive coverage across multiple dimensions: (1) **linguistic diversity**: including both high-resource languages (English, Arabic, Russian) and lower-resource languages (Norwegian Bokmål, Danish) [1] to test cross-lingual generalization, (2) **domain variety**: spanning news, social media, encyclopedic content, scientific literature, and forum discussions to capture real-world application diversity, (3) **construction methods**: including both curated human annotations and synthetic dataset creation to understand how dataset origin affects human-model alignment, (4) **task relevance**: using tasks from established benchmarks widely adopted in the embedding evaluation community, and (5) **task complexity variation**: ranging from binary classification to fine-grained similarity judgments. This systematic selection ensures our findings generalize across the diverse landscape of embedding applications while maintaining direct relevance to existing evaluation frameworks.

Each task category uses a primary evaluation metric to enable consistent human–model comparisons. We summarize the datasets, their domains, and the primary metrics applied in Appendix A. Detailed task examples are provided in Appendix C.

**Retrieval Proxy via Reranking**   We use Reranking as a human-evaluable proxy for Information Retrieval. Direct human evaluation of large-scale retrieval is methodologically infeasible—requiring annotators to evaluate thousands of candidate documents per query. Reranking preserves the core semantic challenge of discriminating query-candidate relevance while remaining tractable: humans evaluate only the top-k candidates, establishing a baseline conceptually equivalent to embedding-space behavior and ensuring human-model comparability.

### 3.3 INSTRUCTIONS

Human instructions are designed to match the task definitions exactly (e.g., identical label sets for classification, same 1-5 scale for STS) to ensure valid comparisons. However, formal, detailed annotation protocols are not publicly available for many MTEB datasets, which limits our ability to verify alignment. To mitigate this, we designed instructions based on the original dataset papers'

---

[1] With 0 being "The Left-Behinds" and 5 being "The Winners", we cover eng: 5, ara: 5, rus: 4, dan: 3, nob: 1 according to the 0-5 scale by (Joshi et al., 2021).

task descriptions, ensuring annotators understood the semantic distinctions required for each task. Instructions were piloted with a small subset before full annotation to identify and resolve ambiguities.

## 3.4 Annotation Procedure

Our annotations process follows a trend similar to recent embedding benchmarks Enevoldsen et al. (2025); Xiao et al. (2025) focusing on a diverse set of tasks with fewer samples rather than large singular tasks. We choose this approach as it allow us to better cover the broad scopes of current benchmark. Annotators were recruited with a focus on cultural and language diversity.

All annotations are conducted in Argilla (Argilla Project Contributors, 2025) using task-specific interfaces: binary relevance for reranking, categorical labels for classification, free cluster ID assignment for clustering, and 0–5 similarity scores for STS. Sample sizes balance task complexity: reranking (20–49 queries), classification (40–48 examples), clustering (30 items), and STS (30–50 pairs).

All annotators were male, aged 20–35, from culturally diverse backgrounds, and experienced NLP practitioners with native or near-native proficiency in the evaluated languages. They followed structured guidelines and completed all annotations independently, without access to ground truth or model predictions. The downsampled task subsets used for comparisons are included in the MTEB package, with detailed task examples provided in Appendix C.

English tasks were annotated by two annotators to enable agreement analysis. Multilingual tasks were annotated by a single annotator with corresponding language expertise. Inter-annotator agreement was assessed with task-appropriate metrics: Fleiss' kappa (Fleiss, 1971) for classification, pairwise ARI (Strehl & Ghosh, 2003) for clustering, pairwise Spearman correlation (Agirre et al., 2012a) for STS, and mean Spearman/Kendall's tau (Manning et al., 2008) for reranking. A detailed agreement analysis is provided in Appendix F. This controlled evaluation setup minimizes potential confounds from dataset variation and enables direct performance comparisons on identical evaluation instances.

## 3.5 Model Selection and Evaluation

We evaluate 13 embedding models chosen to cover multiple dimensions: (1) **parameter scale** (22M–7B), (2) **architecture** (encoder- and decoder-based), (3) **instruction tuning** (instruction-tuned and standard), and (4) **multilingual capability** (English and multilingual). This selection spans diverse computational budgets and training paradigms, capturing the current embedding landscape. All evaluated models are provided in Appendix H.

All models are evaluated on the downsampled instances annotated by humans, using identical metrics, protocols, and computational settings. Human performance is computed using the metrics in Appendix A, mirroring MTEB protocols. For primary analyses, we report MAP for reranking, Accuracy for classification, V-Measure for clustering, and Spearman correlation for STS.

To account for sample size constraints, we determine statistical significance using 95% confidence intervals computed via Wilson Score Intervals (accuracy) and Fisher $z$-transformations (correlation), as detailed in Appendix E.

## 3.6 LLM-as-Annotator Evaluation

To assess whether automated evaluation can proxy human judgment, we evaluate nine state-of-the-art Large Language Models (LLMs) as annotators on the exact same tasks. We employ GPT-5 (full and mini), GPT-4.1 (full and mini), Gemini 2.5 Flash, Mistral Small-24B-Instruct, and three Qwen3 variants (30B, 32B, Coder-30B), prompting them with identical instructions provided to human annotators (see Appendix C).

LLMs receive the same task instances, evaluation metrics, and scoring protocols as human annotators, enabling direct performance comparisons. For classification and STS tasks, LLMs provide categorical labels or numerical similarity scores. For reranking, LLMs rank candidate documents by relevance to the query. Clustering tasks were excluded from LLM evaluation due to fundamental difficulties in eliciting consistent, structured cluster assignments from generative models.

This controlled setup determines whether LLMs can serve as scalable, low-cost proxies for human evaluation or whether they exhibit systematic biases that limit their utility for benchmark development.

By evaluating LLMs on the same instances as humans and embedding models, we can directly compare their annotation quality and identify task-specific strengths and limitations.

# 4 RESULTS AND ANALYSIS

Figure 1 provides an overview of human performance relative to 13 state-of-the-art embedding models across 16 tasks. Human annotators rank 4th overall with a score of 0.776, trailing 3 large models but outperforming 10 others. However, a raw ranking obscures the nuance of task difficulty and data reliability. As shown in Table 1, humans neither represent a uniform performance ceiling nor a lower bound, but rather occupy a middle ground that varies significantly by task category, language, and dataset quality.

We computed 95% confidence intervals for human performance using metric-appropriate methods (Wilson Score Intervals for classification accuracy, Fisher $z$-transformation for correlation-based metrics, and empirical annotator ranges for clustering and reranking). Models perform outside human CIs in 14 of 26 tasks ($p < 0.05$), often on datasets with low inter-annotator agreement where "superhuman" performance may reflect artifact fitting rather than genuine capability (see Appendix E for methodology and complete results). Below we analyze performance patterns by task category and language, with full per-task results in Appendix B.

| Model | Classification | | | | Clustering | | | | Reranking | | | STS | | | Overall |
|---|---|---|---|---|---|---|---|---|---|---|---|---|---|---|---|
| | ara | eng | nob | rus | ara | dan | eng | rus | dan | eng | nob | ara | eng | rus | |
| Number of datasets | (1) | (4) | (1) | (1) | (1) | (1) | (4) | (1) | (1) | (4) | (1) | (1) | (4) | (1) | (26) |
| all-MiniLM-L6-v2 | 57.2 | 58.8 | 51.7 | 55.5 | 35.2 | 24.5 | 55.1 | 31.4 | 78.4 | 93.7 | 71.2 | 6.2 | 83.5 | 33.1 | 61.9 |
| all-mpnet-base-v2 | 53.5 | 62.0 | 47.0 | 60.5 | 21.4 | 22.9 | 59.7 | 36.9 | 79.0 | 93.3 | 80.5 | 13.2 | 83.0 | 42.2 | 63.4 |
| e5-mistral-7b-instruct | 74.5 | 70.0 | 70.5 | 70.0 | 68.5 | **76.0** | 82.7 | **77.7** | 90.6 | **96.4** | 86.1 | 16.0 | 85.9 | 63.0 | 78.2 |
| embeddinggemma-300m | 71.0 | 58.6 | 54.0 | 73.5 | 19.2 | 43.7 | 65.1 | 36.9 | 74.3 | 86.9 | 71.4 | 36.7 | 69.9 | 66.2 | 64.2 |
| gte-Qwen2-1.5B-instruct | 75.2 | 76.5 | 70.8 | 74.5 | 73.7 | 67.1 | 75.9 | 72.2 | 84.3 | 95.3 | 87.8 | 28.8 | 84.0 | 54.2 | 77.5 |
| jasper_en_vision_language_v1 | 63.5 | **87.1** | 70.5 | 79.8 | 64.3 | 54.7 | 83.2 | 43.7 | 90.1 | 95.8 | 90.0 | 40.9 | 88.1 | **69.5** | **80.1** |
| multilingual-e5-base | 75.8 | 64.7 | 73.8 | 77.2 | 35.9 | 40.6 | 45.6 | 36.2 | 92.2 | 94.4 | 87.5 | 31.0 | 85.2 | 62.7 | 68.2 |
| multilingual-e5-large | 77.0 | 64.9 | 75.0 | 80.0 | 34.6 | 31.0 | 52.5 | 46.9 | 95.0 | 95.3 | 92.2 | 33.8 | 86.3 | 68.8 | 70.4 |
| multilingual-e5-small | 72.2 | 62.2 | 69.2 | 81.2 | 35.5 | 38.0 | 51.7 | 59.1 | 88.6 | 94.2 | 88.3 | 28.8 | 85.2 | 60.3 | 69.0 |
| mxbai-embed-large-v1 | 57.2 | 66.4 | 52.2 | 59.0 | 26.5 | 34.2 | 61.9 | 30.5 | 90.8 | 94.5 | 82.0 | 12.7 | 87.6 | 43.7 | 66.6 |
| Qwen3-Embedding-0.6B | 77.2 | 74.7 | 59.8 | 74.8 | 78.8 | 58.5 | 68.4 | 68.3 | 90.0 | 95.5 | 83.6 | 38.0 | **88.5** | 60.3 | 76.8 |
| SFR-Embedding-Mistral | 77.5 | 69.8 | 68.8 | 72.5 | 73.1 | 71.2 | 85.1 | 68.9 | 89.2 | 96.3 | 86.1 | 15.3 | 86.4 | 64.0 | 78.3 |
| stella_en_1.5B_v5 | 65.8 | 84.0 | 67.0 | 79.2 | 36.8 | 42.6 | 78.6 | 46.7 | 91.7 | 96.0 | 88.6 | 37.2 | 86.7 | 62.1 | 76.9 |
| Human | **95.0** | 70.3 | **85.0** | **92.5** | 76.0 | 62.7 | 67.4 | 68.0 | 91.4 | 87.2 | 89.8 | **67.5** | 83.1 | 58.7 | 77.6 |

Table 1: Human performance compared to 13 embedding models across task categories and languages. **Bold** indicates highest performance (human or model), underline indicates best model performance. Humans achieve top performance in 5 of 14 *aggregated* task-language pairs, particularly excelling in non-English sentiment analysis and Arabic semantic similarity. Overall results are aggregated over the 26 task-language pairs.

## 4.1 PERFORMANCE PATTERNS BY TASK CATEGORY

Figure 2 shows human performance relative to model performance across all 26 task-language pairs. Each task shows human performance (point) positioned within the full spectrum from worst to best model performance (range bars). Humans consistently perform in the upper portion of model ranges, typically exceeding median model performance (61.5% of tasks) while rarely matching the best models (15.4% of tasks). Classification tasks show the strongest human performance, with humans outperforming all models in 3 of 7 tasks, while clustering and reranking reveal consistent gaps where humans fall short of top-performing models. Detailed gap analysis can be found in Appendix G.

**Classification:** Human performance averages 70.3, ranging from 45.8 on emotion classification ($\kappa = 0.39$, fair agreement) to 95.0 on Arabic sentiment analysis. Models generally exceed human performance (best: 87.1), but humans outperform models on non-English sentiment analysis, particularly in Arabic (95.0 vs. 77.5) and Russian (92.5 vs. 81.2), likely benefiting from native cultural and linguistic understanding that current models fail to capture.

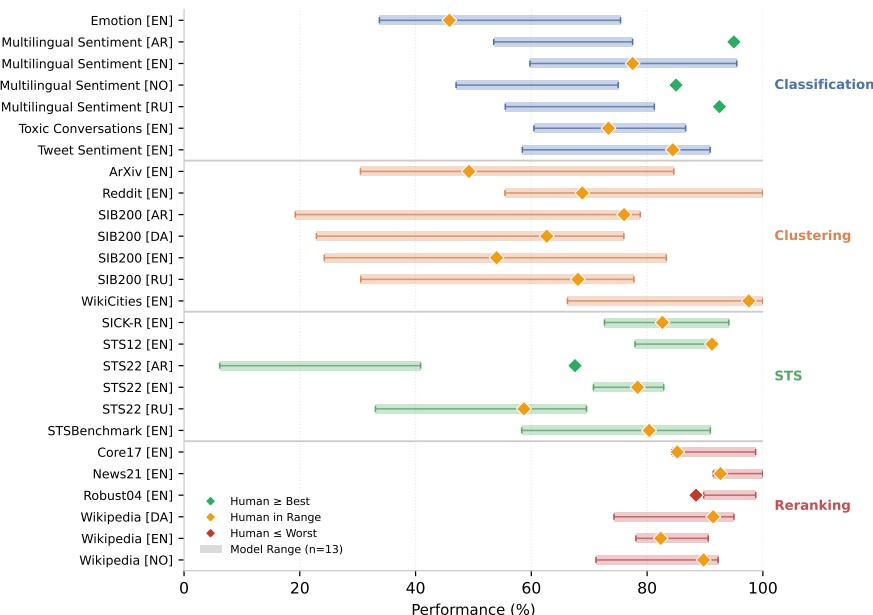

Figure 2: Comprehensive view of human performance relative to all model performance ranges across 16 tasks by language.

**Clustering:** Humans average 67.4 V-measure with extreme variation. Near-perfect performance on WikiCities (97.6, ARI = 0.91) contrasts sharply with poor performance on ArXiv papers (49.2, ARI = −0.001). Models consistently outperform humans (best: 85.1%). The poor inter-annotator agreement on ArXiv indicates fundamental task ambiguity rather than human limitation.

**Reranking:** Humans achieve strong performance (87.2 average MAP) with high inter-annotator agreement ($\rho = 0.64$-$0.85$), demonstrating intuitive relevance understanding. Models exceed human performance (best: 96.4), but the high human agreement suggests these tasks align well with human judgment and provide reliable evaluation targets.

**STS:** Humans average 83.2 Spearman correlation, with notable variation: STS12 achieves 91.2 while STS22-Russian drops to 58.7, likely reflecting dataset quality issues discussed in §4.3. Models achieve comparable performance (best: 88.5), with moderate inter-annotator agreement ($\rho = 0.58$-$0.77$).

These results reveal a critical insight that challenges conventional evaluation paradigms: human performance variation often reflects task quality rather than human limitations. Tasks with high human performance and agreement (reranking, toxicity classification) represent well-specified problems with clear ground truth, while tasks with low human agreement (emotion classification, academic clustering) may suffer from ambiguous annotation guidelines or inherently subjective judgments. Models achieve "superhuman" performance by reproducing consistent label patterns from training data, but this consistency may mask fundamental issues with task specification. Rather than treating low human performance as a ceiling to surpass, our findings suggest that it often signals the need for improved task design and clearer annotation frameworks.

## 4.2 CROSS-LINGUAL PERFORMANCE ANALYSIS

Human win rates vary substantially by language and comparison baseline (see Appendix G for detailed breakdown). Against the best models, humans win only 15% of tasks overall, but this rises to 62% against median models. The advantage is strongest for non-English tasks: humans achieve a 29% win rate on multilingual tasks versus 0% on English-only tasks against best models.

Arabic exhibits the strongest human advantage: a 67% win rate against the best models and 100% against the mean, with the largest gap in semantic similarity (67.5% vs. 40.9%, a 26.6-point margin).

Russian and Norwegian also show consistent human superiority in sentiment analysis, where humans achieve 92.5% and 85.0%, respectively, substantially outperforming the best models. These advantages likely stem from cultural and contextual knowledge that models fail to capture, especially in lower-resource languages.

English tasks are more balanced, with models generally matching or exceeding humans, reflecting the language's dominance in training data. Danish shows mixed outcomes, possibly due to stronger multilingual coverage for Germanic languages.

### 4.3 DATASET QUALITY AND EVALUATION CHALLENGES

Our analysis reveals systematic quality issues in several MTEB datasets that fundamentally compromise their reliability as evaluation benchmarks. Human performance variation often correlates with underlying ambiguity rather than genuine human limitations, providing a diagnostic tool for identifying problematic evaluation targets. See Appendix D for further qualitative analysis of these failure modes; below, we highlight two such examples.:

**Emotion Classification Ambiguity:** The emotion classification dataset exemplifies inherent labeling ambiguity, achieving only fair inter-annotator agreement ($\kappa = 0.39$) with 52.1% consensus. Real examples demonstrate the fundamental challenges: "I feel like such a noob when the customers make really dull and stupid jokes that im supposed to find funny" could reasonably be labeled as sadness (0), anger (3), or even surprise (5) depending on interpretation. Similarly, "I am feeling very indecisive and spontaneous" contains mixed emotional states that resist single-label categorization. Sarcastic expressions like "I got paid too much because I get so many deliveries at work Im feeling a bit shamed" present surface emotions that differ from intended meaning. When human experts fundamentally disagree on correct answers for such inherently ambiguous cases, the apparent "superhuman" model performance (87.1% vs. 45.8% human) likely reflects consistent reproduction of arbitrary majority label patterns rather than superior emotional understanding.

**ArXiv Clustering Breakdown:** Academic paper clustering shows complete breakdown of human agreement (ARI=-0.001), indicating fundamental disagreement about how to categorize academic papers. Real examples illustrate the core ambiguity: papers like "Self-Supervised Audio-Visual Representation Learning with Relaxed Cross-Modal Synchronicity" could legitimately cluster with computer vision, machine learning, or audio processing groups depending on the annotator's perspective on primary methodology versus application domain. "The architecture of innovation: Tracking face-to-face interactions with ubicomp technologies" spans social science, computer science, and architecture domains. Such interdisciplinary papers create fundamental disagreement about correct clustering approaches, with no objectively correct answer. The task uses derived labels from ArXiv categories, but the core issue is that academic papers often span multiple domains, making any single clustering scheme inherently ambiguous. The high model performance (84.6% vs. 49.2% human) suggests that the models are reproducing consistent labeling patterns rather than solving the fundamental categorization challenge.

**High-Quality Benchmark Identification:** Conversely, tasks with high human agreement provide reliable evaluation targets. Reranking tasks achieve strong inter-annotator agreement ($\rho = 0.64-0.85$) with clear performance targets, while toxicity classification shows moderate agreement ($\kappa = 0.55$) with 77.8% annotator consensus. These represent genuine evaluation challenges where model improvements likely reflect meaningful progress rather than pattern matching to flawed labels.

These patterns suggest that apparent "superhuman" model performance often occurs precisely where human agreement is lowest, indicating that models excel not through superior understanding but through consistent reproduction of systematic labeling patterns. This raises concerns about the label quality in embeddings benchmarks, and we encourage future benchmark developers to critically examine the datasets before including them in a benchmark, potentially using human annotations framework like HUME. Detailed analysis of specific quality issues is provided in Appendix D.

### 4.4 CAN LLMS REPLACE HUMAN ANNOTATORS?

We benchmark nine LLMs as annotators to assess whether they can serve as reliable proxies for human judgment. As shown in Table 2, the best-performing LLM (GPT-4.1-mini) achieves 76.1%

| Task | Human | GPT-5 | | GPT-4.1 | | Gemini | Mistral | Qwen3 | | | Best Emb. |
| | | Full | Mini | Full | Mini | 2.5 Flash | Small-24B-I | 30B | 32B | Coder | Score |
|---|---|---|---|---|---|---|---|---|---|---|---|
| Classification | 79.1 | **78.9** | 77.2 | 76.6 | 76.1 | 77.6 | 73.8 | 74.2 | 73.0 | 76.3 | 80.3 (jasper) |
| Reranking | 88.3 | 75.1 | 75.5 | 75.7 | 77.2 | 76.2 | **78.0** | 75.6 | 74.8 | 73.8 | 94.8 (e5) |
| STS | 76.5 | 73.0 | 69.0 | 74.9 | 74.9 | 69.3 | **75.0** | 67.1 | 68.6 | 71.3 | 77.1 (jasper) |
| **Average** | 81.2 | 75.8 | 74.1 | 75.8 | **76.1** | 74.5 | 75.5 | 72.4 | 72.2 | 73.9 | – |

Table 2: LLM-as-annotator performance compared to human annotators and best embedding models per task category. Human and LLM performance is computed over 19 task-language pairs (clustering tasks excluded due to difficulty eliciting cluster assignments). Best embedding model per task category shown with abbreviated name: jasper (jasper_en_vision_language_v1), SFR (SFR-Embedding-Mistral), e5 (multilingual-e5-large). **Bold** indicates best LLM performance (humans and embedding models consistently outperform LLMs and are not bolded).

average accuracy, falling short of human performance (81.2%). We exclude clustering tasks in this comparison due to the difficulty of eliciting cluster assignments from generative models.

Task-specific patterns reveal important limitations. On classification, LLMs approach human performance (GPT-5: 78.9% vs. Human: 79.1%). However, a substantial gap emerges in reranking, where humans achieve 88.3% compared to the best LLM at 78.0% (Mistral-Small): a 10-point deficit on tasks where humans show high agreement ($\rho = 0.64$–$0.85$). For STS, humans (76.5%) outperform all LLMs (best: 75.0%, Mistral-Small). Embedding models achieve the highest scores across all categories (Classification: 80.3%, Clustering: 79.1%, Reranking: 94.8%, STS: 77.1%). Detailed per-task LLM performance is provided in Appendix I.

To assess whether humans and LLMs face similar challenges, we computed Spearman rank correlations between human and LLM performance across the 19 task-language pairs. The moderate positive correlation ($\rho = 0.52$, $p < 0.05$, $n = 19$) indicates that tasks where humans perform well also tend to be tasks where LLMs perform well, suggesting partially shared difficulty patterns. However, the correlation is moderate rather than strong, indicating that humans and LLMs do not face identical challenges across all tasks (see § I.1 for detailed correlation analysis).

## 5 DISCUSSION

### 5.1 TASK QUALITY AND EMBEDDING EVALUATION RELIABILITY

Our analysis reveals a striking pattern: models achieve their highest relative performance precisely where human experts show the least agreement. This confirms that on low-quality datasets, current metrics do not measure semantic understanding but rather the model's ability to reproduce consistent annotation artifacts.

This finding reframes the role of human evaluation in benchmark design. Rather than serving merely as a performance benchmark, human consensus establishes a *validity threshold* for evaluation tasks. When models significantly exceed this bound on low-agreement tasks, it signals that the benchmark itself has lost its descriptive power: the task may be measuring annotation artifacts rather than the capability it claims to assess. HUME provides the empirical mechanism to identify and deprecate these invalid evaluation targets, ensuring that leaderboards measure genuine capability rather than overfitting to noise.

Tasks with high human agreement, such as reranking and toxicity classification, provide reliable evaluation targets. Conversely, low-agreement tasks (e.g., ArXiv clustering, Emotion classification) suffer from ambiguous guidelines or subjective judgments. Cultural factors add another dimension to evaluation reliability. Humans retain substantial advantages in Arabic semantic similarity and multilingual sentiment analysis, revealing genuine model limitations in cross-cultural understanding.

These findings suggest reliable evaluation depends as much on task quality as model capability. Rather than treating high model performance as automatic progress, we recommend prioritizing high-agreement tasks for development, addressing cultural competence gaps, and critically examining whether apparent model superiority on ambiguous tasks reflects genuine capability or evaluation

bias. Novel benchmarks should report human agreement alongside model scores: 85% accuracy on emotion classification (185% of human performance, $\kappa = 0.39$) represents a fundamentally different achievement than 85% on reranking (97% of human performance, $\rho = 0.75$).

## 5.2 LLM-BASED EVALUATION

Our LLM annotation experiments reveal important limitations for using LLMs as proxies for human judgment. While LLMs achieve competitive performance on classification tasks, a substantial gap emerges on reranking, where humans significantly outperform even the best LLMs. Notably, reranking tasks show strong inter-annotator agreement ($\rho = 0.64$–$0.85$), suggesting LLMs struggle with precisely the nuanced relevance judgments where human consensus is highest. This pattern contrasts with classification, where lower human agreement ($\kappa = 0.24$–$0.55$) coincides with near-parity between humans and LLMs.

This has important implications for benchmark development. The reranking gap suggests that LLMs may not reliably capture the semantic distinctions humans make on well-defined tasks, even as they approach human performance on more ambiguous ones. Using LLMs for large-scale annotation may therefore introduce systematic biases, particularly for tasks requiring fine-grained semantic judgments. The architectural mismatch between generative LLMs and discriminative evaluation tasks further limits their utility, as evidenced by our inability to evaluate clustering tasks.

While LLMs offer scalability advantages, these limitations suggest they should augment rather than replace human annotation, particularly for benchmark development where task quality directly impacts model development priorities. Future work should explore hybrid approaches that leverage LLM efficiency for initial annotation while reserving human judgment for high-agreement tasks and uncertain cases.

## 5.3 IMPLICATIONS FOR MODEL DEVELOPMENT AND EVALUATION PRACTICES

Our findings reveal concrete directions for both embedding model development and evaluation methodology that address the fundamental quality issues we've identified.

**Prioritize High-Agreement Tasks for Development:** Development efforts should focus on tasks where humans demonstrate both high performance and agreement, as these provide the most reliable benchmarks for measuring genuine progress. Reranking tasks, with their clear performance targets and strong agreement ($\rho = 0.64 - 0.85$), offer dependable evaluation where the persistent model-human gap (96.4% vs. 87.2%) represents meaningful challenges requiring better modeling of relevance relationships. Toxicity classification, despite moderate agreement ($\kappa = 0.55$), provides another reliable target with 77.8% human consensus. In contrast, optimizing for tasks with poor human agreement (emotion classification $\kappa = 0.39$, ArXiv clustering $ARI = -0.001$) may lead models to excel at reproducing arbitrary labeling patterns rather than developing genuine semantic capabilities.

**Address Cultural and Linguistic Competence Gaps:** The substantial human advantages in non-English tasks reveal critical model limitations that scaling training data alone cannot address. Arabic semantic similarity shows the largest human advantage (67.5% vs. 40.9% best model), while multilingual sentiment demonstrates consistent human superiority in non-English languages (95.0% Arabic, 92.5% Russian). These gaps suggest that current models lack the cultural and contextual knowledge necessary for cross-lingual understanding, requiring architectural innovations or training approaches that go beyond simple data scaling to capture cultural nuances and contextual understanding.

**Replace Problematic Benchmark Datasets:** Our analysis identifies specific datasets that compromise benchmark reliability and should be replaced in future MTEB iterations: emotion classification , ArXiv clustering, and STS22-Russian (systematic parsing artifacts). These tasks provide unreliable evaluation targets that may mislead model development efforts by rewarding pattern matching to flawed gold standards. Replacement datasets should demonstrate reasonable human agreement and clear task specifications, validated through human evaluation before inclusion in benchmark suites.

**Report Dataset Quality Measures:** Model performance should be interpreted in light of dataset quality indicators to provide proper context for evaluation results. We propose that benchmark leaderboards report human agreement metrics alongside model scores. A model achieving 85% accuracy

on emotion classification (185% of human performance, $\kappa = 0.39$) represents a fundamentally different achievement than 85% on reranking (97% of human performance, $\rho = 0.75$). High model performance on low-agreement tasks should be viewed skeptically as potential artifacts of flawed evaluation targets rather than genuine capability improvements. For tasks where human agreement falls below established thresholds ($\kappa < 0.4$ or $\rho < 0.6$), we recommend either improving task specifications or removing the dataset from benchmark suites entirely. However, it is important to recognize that some degree of human disagreement reflects natural variability in judgment rather than dataset flaws. Future benchmarks could incorporate evaluation frameworks that preserve and leverage this variability rather than collapsing it to single gold labels (Plank, 2022; Basile et al., 2021).

### 5.4 Limitations

Our study has several limitations. First, our prioritization of breadth over depth—covering 16 diverse tasks—resulted in smaller sample sizes per task (20–50 instances). While we provide significance analyses to validate our statistical conclusions, larger samples would better capture the full complexity of human performance variation and provide more robust estimates of human judgment distributions.

Second, our multi-task design constrained task-specific training. Annotators were average or above-average raters without specialized training; experts would likely perform better, particularly on technical tasks. This was compounded by sparse annotation guidelines in original datasets, making alignment with original procedures difficult—though this reflects realistic annotation scenarios where perfect replication is often infeasible.

Third, while three annotators participated overall, only two annotations were collected for most tasks, limiting our ability to fully characterize agreement patterns. Additionally, while we ensured cultural and linguistic diversity among annotators, they were all male and aged 20–35, which does not fully represent human judgment distributions across broader demographic groups.

Fourth, While our study quantifies where models diverge from human performance, it does not fully explain why these gaps arise. Identifying the underlying factors - such as gaps in training data coverage, domain or cultural biases, and linguistic variability, particularly in low-resource settings - remains a critical direction for future research. However, detailed information about model training corpora is often unavailable, limiting such analysis.

Finally, while our study evaluates human performance across diverse tasks, we did not systematically investigate how task design features—such as specification clarity versus meaningful challenge—affect human agreement and model discrimination. The field needs rigorous research on these design principles to avoid both ambiguity, which depresses human agreement, and oversimplification, which diminishes discriminative power.

## 6 Conclusion

We introduce HUME, a comprehensive human evaluation framework for MTEB, addressing a critical gap in understanding empirical performance bounds for embedding models. By measuring human performance across 16 datasets spanning reranking, classification, clustering, and STS, we establish statistically robust baselines that reframe how model achievements should be interpreted.

Our findings show that human performance varies substantially by task categories. Tasks with high agreement provide reliable benchmarks, while low-agreement tasks often reveals design issues in the task formulation.

Finally, our benchmarking of nine LLM-as-annotator systems demonstrates that while they offer scalability, they cannot yet replace human judgment entirely. The best LLM (GPT-4.1-mini, 76.1%) falls short of human performance (81.2%), particularly on reranking. This suggests that future benchmarks should leverage LLMs to augment, but not replace, human evaluation.

### 6.1 Ethical considerations

Annotators were co-authors who consented to the study. There were no external crowd workers involved in any part of the annotation process.

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

# A   DATASET SPECIFICATIONS

Table 3 details the 16 datasets selected for this study, including their domains, descriptions, and the primary metrics used for evaluation.

| | Datasets | Description | Metrics |
|---|---|---|---|
| Reranking | Core17, News21, Robust04 (Weller et al., 2024) | Information retrieval benchmarks (news, documents) | MAP, MRR@10, nDCG@10 |
| | WikiMulti (Enevoldsen et al., 2025) | Wikipedia article reranking (eng, dan, nob) | |
| Classification | Emotion (Saravia et al., 2018) | Emotion classification from social media text | Accuracy, F1, Weighted F1 |
| | Tweet Senti (Barbieri et al., 2022) | Sentiment analysis of tweets | |
| | Toxicity (cjadams et al., 2019) | Toxic content detection | |
| | Multilingual Sentiment (Mollanorozy et al., 2023) | Sentiment classification (ara, eng, nob, rus) | |
| Clustering | WikiCities (Foundation) | Entity clustering from Wikipedia | V-Measure, ARI, AMI |
| | ArXiv (arXiv.org submitters, 2024) | Academic paper topic clustering (derived labels) | |
| | Reddit (Geigle et al., 2021) | Forum discussion topic clustering | |
| | SIB200 (Adelani et al., 2023) | Multilingual sentence clustering (ara, dan, eng, rus) | |
| STS | STSBenchmark (May, 2021) | General semantic similarity benchmark | Spearman, Pearson |
| | SICK-R (Marelli et al., 2014) | Semantic relatedness and entailment | |
| | STS12 (Agirre et al., 2012b) | Shared task semantic similarity | |
| | STS22 (Chen et al., 2022) | Multilingual semantic similarity (ara, eng, rus) | |

Table 3: Complete list of 16 datasets and evaluation metrics used for human annotation. For the metrics we use MAP (Manning et al., 2008), MRR (Manning et al., 2008), nDCG (Järvelin & Kekäläinen, 2002), Accuracy/F1 (Sokolova & Lapalme, 2009), V-Measure (Rosenberg & Hirschberg, 2007), ARI (Hubert & Arabie, 1985), AMI (Vinh et al., 2010), Spearman/Pearson (Spearman, 2010) (Pearson, 1895), following the MTEB implementations.

# B   DETAILED RESULTS BY TASK CATEGORY

This section provides comprehensive results for all tasks, organized by category. Each table includes human performance alongside all 13 evaluated models, with inter-annotator agreement metrics where available.

Table 4 presents full results of the clustering tasks. Table 5 presents full results of the classification tasks. Table 6 presents full results of the reranking tasks. Table 7 presents full results of the STS tasks.

# C   TASK EXAMPLES

This section provides screenshots of the actual Argilla annotation interfaces used in our study, illustrating the annotation challenges and interface design that human annotators encountered.

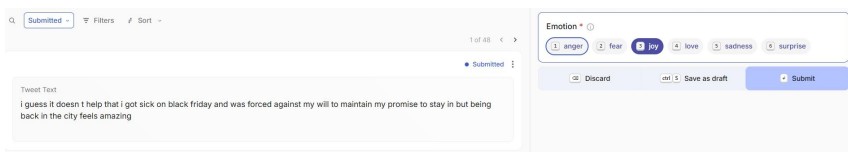

Figure 3: Emotion Classification annotation interface showing the 6-category emotion labeling task. This task achieved fair inter-annotator agreement ($\kappa = 0.39$) due to ambiguous emotional states and mixed emotions in social media text. Human performance: 45.8%, Best model: 87.1%.

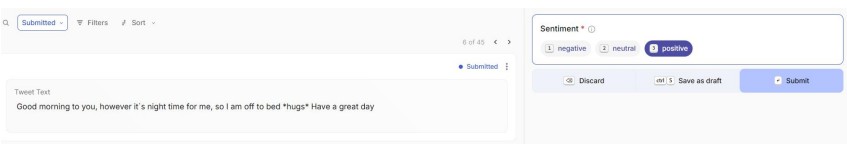

Figure 4: Tweet Sentiment Classification annotation interface demonstrating sentiment polarity annotation. This task achieved moderate inter-annotator agreement ($\kappa = 0.48$) with reasonable consensus on positive/negative sentiment. Human performance: 84.4%, Best model: 90.9%.

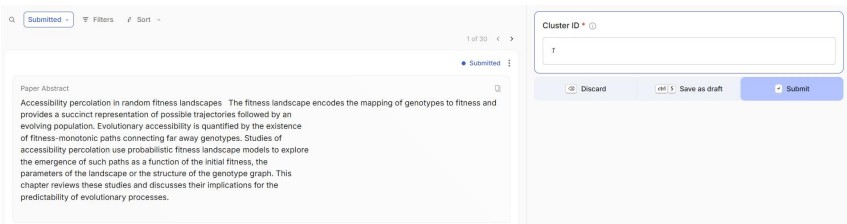

Figure 5: ArXiv Clustering annotation interface showing academic papers that caused complete annotator disagreement (ARI $= -0.001$) due to interdisciplinary research overlap. Papers could be categorized by methodology, application domain, or research community, leading to fundamental disagreement. Human performance: 49.2%, Best model: 84.6%.

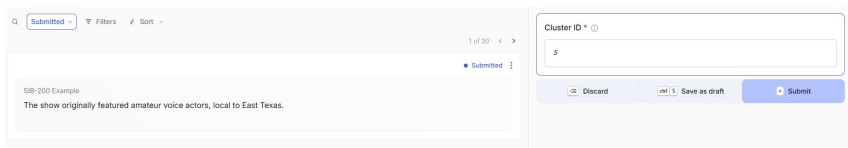

Figure 6: Reddit Clustering annotation interface demonstrating thematic grouping of discussion topics. This task achieved fair agreement (ARI $= 0.34$) due to overlapping themes across different discussion topics. Human performance: 68.8%, Best model: 100%.

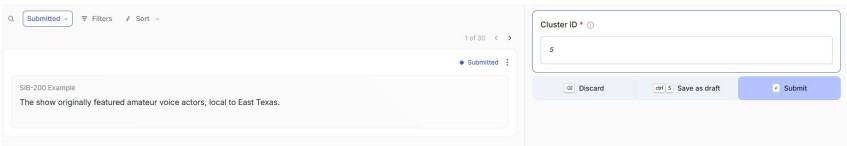

Figure 7: SIB200 Clustering annotation interface showing multilingual sentence clustering task. This task achieved moderate inter-annotator agreement (ARI $= 0.42$) with variation across languages depending on cultural context and sentence complexity.

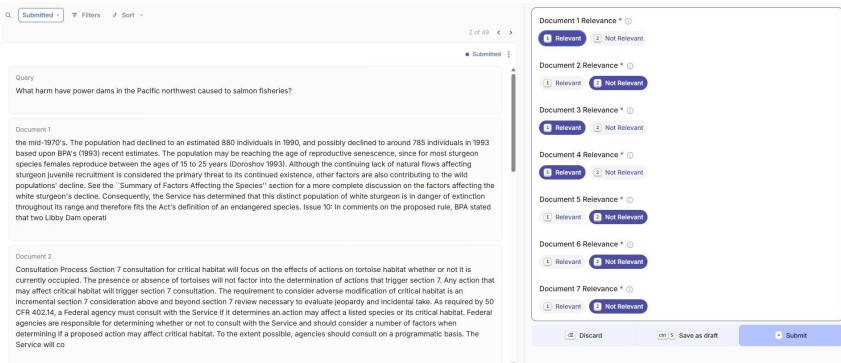

Figure 8: Robust04 Reranking annotation interface showing document relevance assessment for information retrieval queries. This task achieved strong inter-annotator agreement ($\rho = 0.72$). Human performance: 88.5%, Best model: 98.8%.

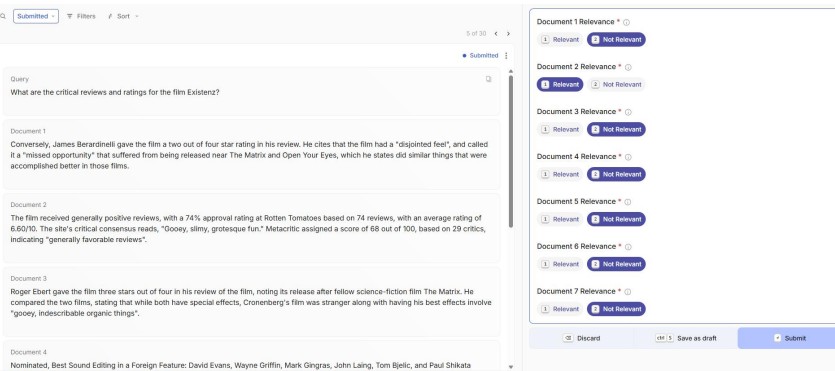

Figure 9: Wikipedia Multilingual Reranking annotation interface demonstrating cross-lingual relevance judgment. This task achieved moderate agreement ($\rho = 0.64$) due to cross-lingual complexity.

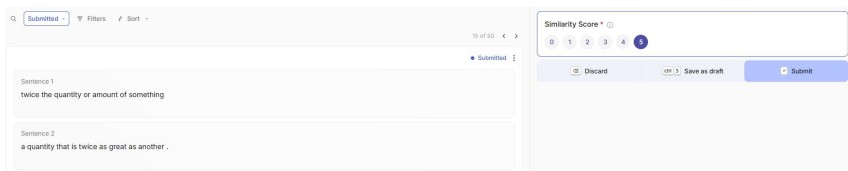

Figure 10: STS12 annotation interface showing semantic similarity assessment using a 0-5 scale. This well-curated dataset achieved strong inter-annotator agreement ($\rho = 0.77$). Human performance: 91.2%, Best model: 92.0%.

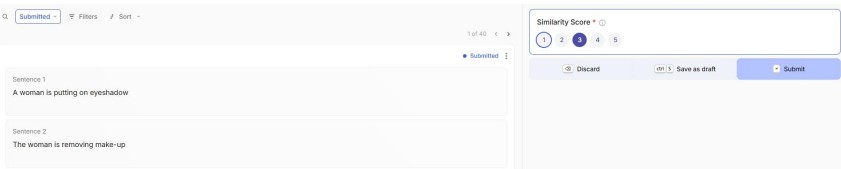

Figure 11: SICK-R annotation interface showing semantic relatedness and entailment task. This task achieved moderate agreement ($\rho = 0.68$) due to task complexity. Human performance: 82.6%, Best model: 94.1%.

| Model | Arxiv | Reddit | SIB200 | WikiCities |
|---|---|---|---|---|
| all-MiniLM-L6-v2 | 61.8 | 56.5 | 33.2 | 73.8 |
| all-mpnet-base-v2 | 56.0 | 63.0 | 33.2 | 86.7 |
| e5-mistral-7b-instruct | 77.2 | 74.9 | 73.8 | 100.0 |
| embeddinggemma-300m | 73.0 | 72.8 | 36.0 | 70.3 |
| gte-Qwen2-1.5B-instruct | 70.8 | 88.4 | 72.3 | 73.6 |
| jasper_en_vision_language_v1 | 83.9 | 95.1 | 59.0 | 95.1 |
| multilingual-e5-base | 31.9 | 60.0 | 34.4 | 66.2 |
| multilingual-e5-large | 51.1 | 55.4 | 37.8 | 69.2 |
| multilingual-e5-small | 30.5 | 76.9 | 38.7 | 72.6 |
| mxbai-embed-large-v1 | 48.3 | 72.1 | 33.0 | 90.8 |
| Qwen3-Embedding-0.6B | 69.1 | 76.3 | 65.5 | 74.8 |
| SFR-Embedding-Mistral | 75.5 | 81.4 | 74.8 | 100.0 |
| stella_en_1.5B_v5 | 84.6 | 100.0 | 44.2 | 86.4 |
| Human | 49.2 | 68.8 | 65.2 | 97.6 |

Table 4: Full clustering results.

| Model | Emotion | MultilSenti | ToxicConvo | TweetSenti |
|---|---|---|---|---|
| all-MiniLM-L6-v2 | 42.1 | 57.9 | 66.4 | 59.6 |
| all-mpnet-base-v2 | 44.0 | 61.5 | 60.4 | 58.4 |
| e5-mistral-7b-instruct | 47.3 | 75.9 | 68.0 | 75.8 |
| embeddinggemma-300m | 40.4 | 64.6 | 66.4 | 67.8 |
| gte-Qwen2-1.5B-instruct | 56.2 | 78.1 | 78.7 | 79.3 |
| jasper_en_vision_language_v1 | 75.4 | 77.3 | 86.7 | 90.9 |
| multilingual-e5-base | 36.2 | 78.6 | 66.0 | 69.1 |
| multilingual-e5-large | 38.5 | 81.1 | 63.3 | 65.3 |
| multilingual-e5-small | 33.8 | 75.8 | 65.1 | 69.6 |
| mxbai-embed-large-v1 | 42.1 | 64.4 | 68.7 | 65.8 |
| Qwen3-Embedding-0.6B | 51.9 | 74.1 | 74.4 | 87.8 |
| SFR-Embedding-Mistral | 46.7 | 77.1 | 67.3 | 75.6 |
| stella_en_1.5B_v5 | 71.9 | 76.1 | 82.9 | 89.1 |
| Human | 45.8 | 87.5 | 73.3 | 84.4 |

Table 5: Full Classification results.

# D  TASK QUALITY ANALYSIS

## D.1  DATASET QUALITY ISSUES

Our analysis revealed quality issues across multiple datasets that significantly impact human-model performance comparisons. These issues fall into several categories that help explain performance patterns observed in our study.

### D.1.1  STS22-RUSSIAN

The Russian subset of "STS22" dataset shows patterns that may help explain the comparatively low human agreement we observed.

| Model | Core17 | News21 | Robust04 | Wikipedia |
|---|---|---|---|---|
| all-MiniLM-L6-v2 | 95.6 | 98.8 | 96.3 | 77.8 |
| all-mpnet-base-v2 | 98.6 | 98.6 | 97.8 | 79.2 |
| e5-mistral-7b-instruct | 98.8 | 99.5 | 98.8 | 88.4 |
| embeddinggemma-300m | 84.2 | 91.4 | 89.8 | 76.0 |
| gte-Qwen2-1.5B-instruct | 97.5 | 99.2 | 98.5 | 86.1 |
| jasper_en_vision_language_v1 | 98.2 | 100.0 | 98.7 | 88.8 |
| multilingual-e5-base | 96.2 | 98.6 | 96.9 | 88.5 |
| multilingual-e5-large | 95.7 | 97.8 | 97.2 | 92.6 |
| multilingual-e5-small | 95.6 | 98.1 | 97.5 | 87.6 |
| mxbai-embed-large-v1 | 97.2 | 98.0 | 98.6 | 85.6 |
| Qwen3-Embedding-0.6B | 97.0 | 100.0 | 98.5 | 86.8 |
| SFR-Embedding-Mistral | 97.9 | 99.7 | 98.8 | 87.9 |
| stella_en_1.5B_v5 | 98.6 | 100.0 | 98.3 | 89.2 |
| Human | 85.2 | 92.7 | 88.5 | 87.9 |

Table 6: Full Reranking results.

| Model | SICK-R | STS12 | STS22 | STSB |
|---|---|---|---|---|
| all-MiniLM-L6-v2 | 91.5 | 85.7 | 48.4 | 81.8 |
| all-mpnet-base-v2 | 89.8 | 83.7 | 54.3 | 78.4 |
| e5-mistral-7b-instruct | 93.2 | 89.1 | 58.5 | 85.9 |
| embeddinggemma-300m | 72.6 | 77.9 | 57.9 | 58.3 |
| gte-Qwen2-1.5B-instruct | 93.4 | 86.9 | 60.3 | 80.0 |
| jasper_en_vision_language_v1 | 93.8 | 92.0 | 67.2 | 88.7 |
| multilingual-e5-base | 91.5 | 86.5 | 63.3 | 82.9 |
| multilingual-e5-large | 89.4 | 89.9 | 65.9 | 83.6 |
| multilingual-e5-small | 88.6 | 87.6 | 63.3 | 81.9 |
| mxbai-embed-large-v1 | 93.4 | 91.1 | 53.9 | 88.9 |
| Qwen3-Embedding-0.6B | 93.3 | 91.8 | 63.6 | 90.9 |
| SFR-Embedding-Mistral | 94.1 | 89.2 | 59.3 | 86.4 |
| stella_en_1.5B_v5 | 92.3 | 89.1 | 64.3 | 87.1 |
| Human | 82.6 | 91.2 | 68.2 | 80.4 |

Table 7: Full STS results.

**Context Expansion Issues:**

- Sentence pairs labeled as "4" (identical meaning) where one sentence contains basic information and the paired sentence includes additional backstory or context

- Translated example pattern: "Company reports earnings" vs. "Company reports earnings of $X million, exceeding expectations due to strong performance in sector Y"

- Human annotators correctly identify these as semantically different (similarity 2-3), while gold labels mark them as identical

- This explains the low human performance on STS22-Russian (58.5%) compared to models (69.5%)

**Parsing and Processing Errors:**

- Incomplete sentence parsing affecting semantic interpretation
- Parsing artifacts from web pages (e.g., page numbers, lists of automatically generated related news, ads)

### D.1.2 MULTILINGUAL SENTIMENT CLASSIFICATION-RUSSIAN

The Russian subset of "MultilingualSentimentClassification" consists of news articles from different news sites. The task is to classify each text as "positive" or "negative". However, the dataset presents several challenges:

**Neutral and Ambiguous Content:**

- Many samples are based on press releases from companies or government departments, which are often neutral in tone and difficult to categorize as positive or negative.
- Translated example: "The total amount of pension savings accumulated in JSC 'Unified Accumulative Pension Fund' (UAPF) as of September 1, 2016, amounted to about 6.41 trillion tenge, the press center of the pension fund said, KazTAG reports. ..."
- Such sentences are more informative than sentiment-bearing.

**Parsing and Processing Errors:**

- Similar to the issues described in § D.1.1, the dataset contains parsing artifacts from web pages.

### D.1.3 EMOTION CLASSIFICATION

The emotion classification dataset suffers from inherent label ambiguity that explains the low human agreement ($\kappa = 0.39$):

**Mixed Emotional States:**

- Texts expressing multiple emotions simultaneously: "i am feeling very indecisive and spontaneous" (labeled as fear but could be surprise)
- "i was feeling pretty anxious all day but my first day at work was a very good day and that helped a lot" (contains both fear and joy)
- "i am feeling crampy and cranky" (physical discomfort mixed with anger)

**Sarcastic and Ironic Expressions:**

- "i got paid too much because i get so many deliveries at work im feeling a bit shamed so will curb the spending for a bit" (sarcasm about being "overpaid")
- "i feel like such a noob when the customers make really dull and stupid jokes that im supposed to find funny" (surface sadness but underlying anger/frustration)
- "i feel very cheated since i am supporting the family and doing all the other stuff while he spends hours a day gaming" (labeled as joy but clearly expressing anger)

**Contextual and Subjective Interpretation:**

- "i feel shame in a strange way" (ambiguous emotional context, labeled as surprise)
- "i feel all funny sometimes" (vague emotional description that could be multiple categories)
- "i feel underappreciated and under valued" (could be sadness, anger, or fear depending on interpretation)

### D.1.4 ARXIV CLUSTERING CHALLENGES

Academic paper clustering presents fundamental categorization difficulties that explain the complete breakdown of human agreement (ARI $= -0.001$). This task uses derived labels from ArXiv paper categories:

**Interdisciplinary Research Papers:**

- "Self-Supervised Audio-Visual Representation Learning with Relaxed Cross-Modal Synchronicity" (could cluster with computer vision, audio processing, or self-supervised learning)
- "The architecture of innovation: Tracking face-to-face interactions with ubicomp technologies" (spans social science, computer science, and architecture)
- "PIINET: A 360-degree Panoramic Image Inpainting Network Using a Cube Map" (computer vision, graphics, or deep learning focus)

**Methodological vs. Application Domain Confusion:**

- "Convergent Actor-Critic Algorithms Under Off-Policy Training and Function Approximation" (reinforcement learning methodology vs. control theory application)
- "Learning-Based Adaptive IRS Control with Limited Feedback Codebooks" (machine learning method vs. wireless communications application)
- "Structure-preserving numerical methods for stochastic Poisson systems" (numerical methods vs. mathematical physics)

**Emerging and Cross-Domain Research:**

- "The modularity of action and perception revisited using control theory and active inference" (cognitive science, control theory, or neuroscience)
- "Food-chain competition influences gene's size" (evolutionary biology, computational biology, or mathematical modeling)
- "Wavelet Analysis of Dengue Incidence and its Correlation with Weather and Vegetation Variables in Costa Rica" (epidemiology, signal processing, or environmental science)

### D.2 IMPACT ON EVALUATION

These quality issues have several critical implications for embedding evaluation:

1. **Artificial Model Advantages:** Models may achieve "superhuman" performance by consistently reproducing systematic labeling patterns rather than demonstrating superior semantic understanding. This is particularly evident in tasks with low human agreement where models can exploit consistent but incorrect labeling patterns.
2. **Misleading Benchmarks:** Tasks with fundamental quality issues provide unreliable targets for model development. High model performance on such tasks may not indicate genuine capability improvements but rather successful pattern matching to flawed gold standards.
3. **Cultural and Linguistic Bias:** Quality issues disproportionately affect non-English tasks, potentially masking genuine model limitations in cross-cultural understanding while artificially inflating performance on problematic English datasets.
4. **Evaluation Validity:** The validity of using these datasets as benchmarks is questionable when human experts cannot agree on correct answers, suggesting fundamental issues with task specification rather than human limitations.

## E  STATISTICAL ROBUSTNESS ANALYSIS

### E.1 CONFIDENCE INTERVAL METHODOLOGY

Given sample size constraints ($N = 20 - 50$), we computed 95% Confidence Intervals (CIs) for human performance using metric-specific analytical methods rather than generic approximations:

- **Classification (Accuracy):** Wilson Score Interval (Wilson, 1927), which is robust for binomial proportions with small sample sizes and avoids the "zero-width" errors of normal approximations.
- **STS (Correlation):** Fisher $z$-transformation (Fisher, 1915) to compute CIs for Spearman correlations, ensuring valid bounds within $[-1, 1]$.
- **Clustering & Reranking:** Empirical range between annotators as a conservative bound given $N_{\text{annotators}} = 2$.

Statistical significance ($^*$) is determined by a non-parametric overlap test: a model is considered significantly different if its score falls outside the human 95% CI (corresponding to $p < 0.05$).

### E.2 DETAILED RESULTS WITH CONFIDENCE INTERVALS

Table 8 presents human performance with 95% confidence intervals for all 26 task-language pairs. Models perform outside the human confidence interval in 14 of 26 tasks, indicating statistically significant differences. Crucially, this separation often occurs in tasks with low to moderate inter-annotator agreement. For example, in EmotionClassification, the best model (75.4) significantly exceeds human performance (45.8, CI:[32.6, 59.7]), but the low agreement ($\kappa = 0.39$) suggests this "superhuman" performance may reflect fitting to annotation artifacts rather than genuine semantic superiority. Conversely, in MultilingualSentiment (Arabic) and STS22 (Arabic), humans significantly outperform models, highlighting genuine cultural gaps that current models fail to bridge.

| Task | Lang | N | K | Tot | Human (95% CI) | IAA | Best Model |
|---|---|---|---|---|---|---|---|
| *Classification* | | | | | | | |
| EmotionClassification | eng | 48 | 2 | 96 | 45.8 [32.6, 59.7] | $\kappa = 0.39$ | 75.4$^*$ |
| MultilingualSentiment | ara | 40 | 1 | 40 | 95.0 [83.5, 98.6] | N/A | 77.5$^*$ |
| MultilingualSentiment | eng | 40 | 2 | 80 | 77.5 [62.5, 87.7] | $\kappa = 0.24$ | 95.5$^*$ |
| MultilingualSentiment | nob | 40 | 1 | 40 | 85.0 [70.9, 92.9] | N/A | 75.0 |
| MultilingualSentiment | rus | 40 | 1 | 40 | 92.5 [80.1, 97.4] | N/A | 81.3 |
| ToxicConversations | eng | 45 | 2 | 90 | 73.3 [59.0, 84.0] | $\kappa = 0.55$ | 86.7$^*$ |
| TweetSentimentExtraction | eng | 45 | 2 | 90 | 84.4 [71.2, 92.3] | $\kappa = 0.41$ | 90.9 |
| *Clustering* | | | | | | | |
| ArxivClusteringP2P | eng | 30 | 2 | 60 | 49.2 [35.3, 63.2] | ARI=-0.00 | 84.6$^*$ |
| RedditClusteringP2P | eng | 30 | 2 | 60 | 68.8 [63.2, 74.4] | ARI=0.42 | 100.0$^*$ |
| SIB200ClusteringS2S | ara | 30 | 1 | 30 | 76.0 [58.4, 87.8] | N/A | 78.8 |
| SIB200ClusteringS2S | dan | 30 | 1 | 30 | 62.7 [44.9, 77.6] | N/A | 76.0 |
| SIB200ClusteringS2S | eng | 30 | 2 | 60 | 54.0 [41.8, 66.3] | ARI=0.15 | 83.3$^*$ |
| SIB200ClusteringS2S | rus | 30 | 1 | 30 | 68.1 [50.2, 81.9] | N/A | 77.7 |
| WikiCitiesClustering | eng | 30 | 2 | 60 | 97.6 [95.2, 100.0] | ARI=0.91 | 100.0 |
| *Reranking* | | | | | | | |
| Core17Instruction | eng | 20 | 2 | 40 | 85.2 [83.6, 86.8] | $\rho = 0.80$ | 98.8$^*$ |
| News21Instruction | eng | 31 | 2 | 62 | 92.7 [91.3, 94.1] | $\rho = 0.85$ | 100.0$^*$ |
| Robust04Instruction | eng | 49 | 2 | 98 | 88.5 [82.2, 94.8] | $\rho = 0.75$ | 98.8$^*$ |
| WikipediaMultilingual | dan | 30 | 1 | 30 | 91.4 [76.2, 97.3] | N/A | 95.0 |
| WikipediaMultilingual | eng | 30 | 2 | 60 | 82.4 [75.6, 89.1] | $\rho = 0.64$ | 90.6$^*$ |
| WikipediaMultilingual | nob | 30 | 1 | 30 | 89.8 [74.1, 96.4] | N/A | 92.3 |
| *STS* | | | | | | | |
| SICK-R | eng | 40 | 2 | 80 | 82.7 [69.4, 90.5] | $\rho = 0.63$ | 94.1$^*$ |
| STS12 | eng | 50 | 2 | 100 | 91.2 [84.9, 94.9] | $\rho = 0.77$ | 92.0 |
| STS22 | ara | 30 | 1 | 30 | 67.6 [41.7, 83.3] | N/A | 40.9$^*$ |
| STS22 | eng | 30 | 2 | 60 | 78.4 [59.0, 89.2] | $\rho = 0.75$ | 82.9 |
| STS22 | rus | 30 | 1 | 30 | 58.7 [28.8, 78.2] | N/A | 69.5 |
| STSBenchmark | eng | 50 | 2 | 100 | 80.4 [67.7, 88.4] | $\rho = 0.58$ | 90.9$^*$ |

Table 8: Human performance with 95% confidence intervals. N = number of samples; K = number of annotators; Tot = total annotations (N×K). CIs computed via Wilson Score Interval (Classification), Fisher's $z$-transformation (STS), and Annotator Range (Clustering/Reranking). $^*$ indicates model score outside human 95% CI ($p < 0.05$). IAA = Inter-Annotator Agreement.

# F  INTER-ANNOTATOR AGREEMENT ANALYSIS

## F.1  AGREEMENT METRICS BY TASK CATEGORY

This section provides detailed inter-annotator agreement analysis using task-appropriate metrics. Agreement levels follow standard guidelines: $\kappa > 0.8$ (excellent), $0.6 < \kappa \leq 0.8$ (substantial), $0.4 < \kappa \leq 0.6$ (moderate), $0.2 < \kappa \leq 0.4$ (fair), $\kappa \leq 0.2$ (poor). For correlations: $\rho > 0.7$ (strong), $0.4 < \rho \leq 0.7$ (moderate), $\rho \leq 0.4$ (weak).

### F.1.1  CLASSIFICATION TASKS

- **Emotion Classification:** $\kappa = 0.39$ (fair agreement)
    - 2 annotators, 48 items, 96 total annotations
    - Mean percentage agreement: 52.1%
    - Performance: Human 45.8%, Best model 87.1%

- **Toxicity Classification:** $\kappa = 0.55$ (moderate agreement)
    - 2 annotators, 45 items, 90 total annotations
    - Mean percentage agreement: 77.8%
    - Performance: Human 73.3%, Best model 86.7%

- **Tweet Sentiment Classification:** $\kappa = 0.41$ (moderate agreement)
    - 2 annotators, 45 items, 90 total annotations
    - Mean percentage agreement: 62.2%
    - Performance: Human 84.4%, Best model 90.9%

- **Multilingual Sentiment Classification:** Agreement only for English
    - English: $\kappa = 0.24$ (fair agreement), 2 annotators, 40 items, 62.5% agreement
    - Arabic, Norwegian, Russian: Single annotator (no agreement metrics)
    - Performance: Human advantages in non-English variants

### F.1.2  CLUSTERING TASKS

- **ArXiv Clustering:** ARI $= -0.001$ (no agreement)
    - 2 annotators, 30 items, 60 total annotations
    - Complete breakdown of consensus on academic paper categories
    - Performance: Human 49.2%, Best model 84.6%

- **Reddit Clustering:** ARI $= 0.42$ (moderate agreement)
    - 2 annotators, 30 items, 60 total annotations
    - Moderate consensus on discussion topic groupings
    - Performance: Human 68.8%, Best model 100%

- **WikiCities Clustering:** ARI $= 0.91$ (excellent agreement)
    - 2 annotators, 30 items, 60 total annotations
    - High consensus on geographical entity groupings
    - Performance: Human 97.6%, Best model 100%

- **SIB200 Clustering:** Agreement only for English
    - English: ARI $= 0.15$ (weak agreement), 2 annotators, 30 items
    - Arabic, Danish, Russian: Single annotator (no agreement metrics)
    - Performance varies significantly across languages

### F.1.3 RERANKING TASKS

- **News21:** $\rho = 0.85$ (strong agreement)
  - 2 annotators, 31 items, 62 total annotations
  - Mean Kendall tau: 0.85, Binary kappa: 0.83
  - Performance: Human 92.7%, Best model 100%

- **Core17:** $\rho = 0.80$ (strong agreement)
  - 2 annotators, 20 items, 40 total annotations
  - Mean Kendall tau: 0.80, Binary kappa: 0.78
  - Performance: Human 85.2%, Best model 98.8%

- **Robust04:** $\rho = 0.75$ (strong agreement)
  - 2 annotators, 49 items, 98 total annotations
  - Mean Kendall tau: 0.75, Binary kappa: 0.72
  - Performance: Human 88.5%, Best model 98.8%

- **Wikipedia Multilingual Reranking:** Agreement only for English
  - English: $\rho = 0.64$ (moderate agreement), 2 annotators, 30 items
  - Mean Kendall tau: 0.64, Binary kappa: 0.60
  - Danish, Norwegian: Single annotator (no agreement metrics)
  - Performance varies across languages

### F.1.4 STS TASKS

- **STS12:** $\rho = 0.77$ (strong agreement)
  - 2 annotators, 50 items, 100 total annotations
  - Performance: Human 91.2%, Best model 92.0%

- **STSBenchmark:** $\rho = 0.58$ (moderate agreement)
  - 2 annotators, 50 items, 100 total annotations
  - Performance: Human 80.4%, Best model 90.9%

- **SICK-R:** $\rho = 0.63$ (moderate agreement)
  - 2 annotators, 40 items, 80 total annotations
  - Performance: Human 82.6%, Best model 94.1%

- **STS22:** Agreement only for English
  - English: $\rho = 0.75$ (strong agreement), 2 annotators, 30 items
  - Arabic, Russian: Single annotator (no agreement metrics)
  - Performance varies significantly by language

### F.2 AGREEMENT PATTERNS AND TASK RELIABILITY

### F.2.1 HIGH-AGREEMENT TASKS (RELIABLE BENCHMARKS)

Tasks with high human agreement ($\kappa > 0.6$ or $\rho > 0.7$) consistently demonstrate:

- Clear, objective task specifications with minimal ambiguity
- Adequate context for making informed judgments
- Cultural and linguistic familiarity for annotators
- Well-defined evaluation criteria with concrete examples
- Minimal dataset quality issues or processing artifacts
- Consistent performance patterns across annotators

**Examples:** WikiCities clustering, News21/Core17/Robust04 reranking, STS12, STSBenchmark

### F.2.2 Low-Agreement Tasks (Problematic Benchmarks)

Tasks with low agreement ($\kappa < 0.4$ or $\rho < 0.6$) often exhibit:

- Ambiguous annotation guidelines or subjective judgment requirements
- Cross-cultural interpretation challenges
- Insufficient context for accurate assessment
- Systematic dataset quality issues or processing artifacts
- Inherently subjective or multi-faceted concepts
- Inconsistent or contradictory gold standard labels

**Examples:** Emotion classification, ArXiv clustering, STS22-Russian

## G    Additional Human vs Model Analysis

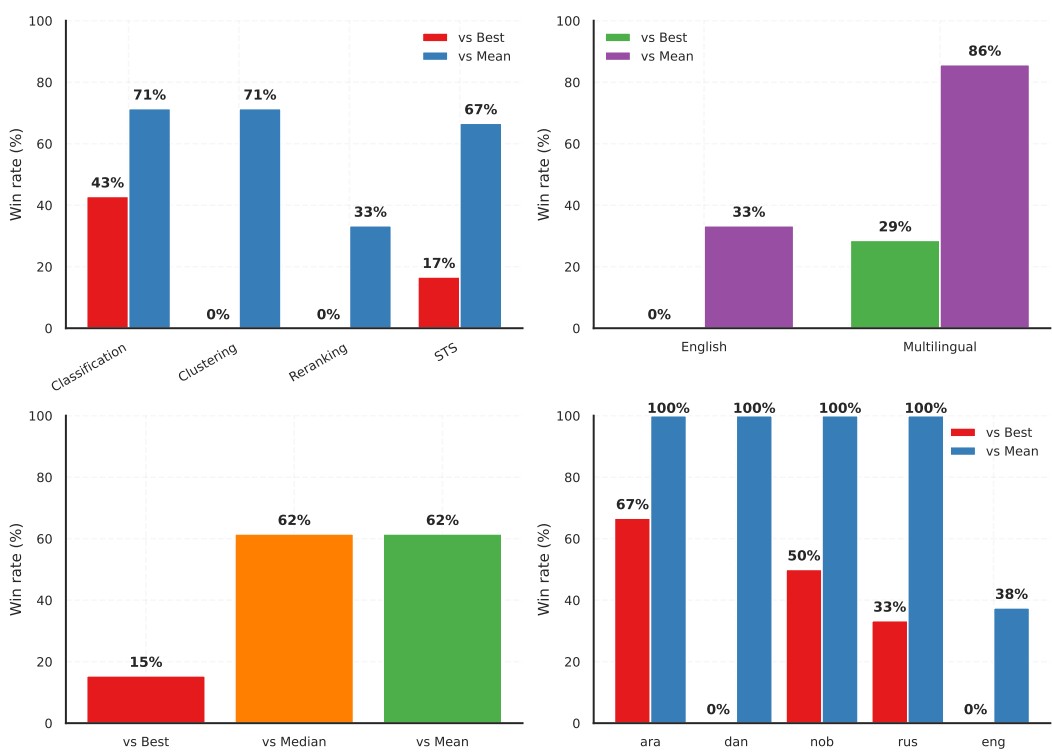

Figure 12: Human win rates across task categories and languages. Top left: By task category shows humans perform moderately in classification but struggle in clustering, reranking, and STS against best models. Top right: English-only vs multilingual tasks reveals humans perform better on multilingual tasks (29% vs 0% against best models). Bottom left: Performance varies dramatically by baseline comparison (15% vs best, 62% vs mean models). Bottom right: Language-specific breakdown shows varying performance across different language codes.

This section contains additional analysis on human vs model. Figure 14 shows the human performance gaps versus median-performing models over all tasks by language. Figure 15 shows the task difficulty categorization based on human performance levels. Figure 16 shows the model consistency analysis showing performance ranges across tasks. Figure 2 shows a comprehensive view of human performance relative to all model performance ranges across 16 tasks by language.

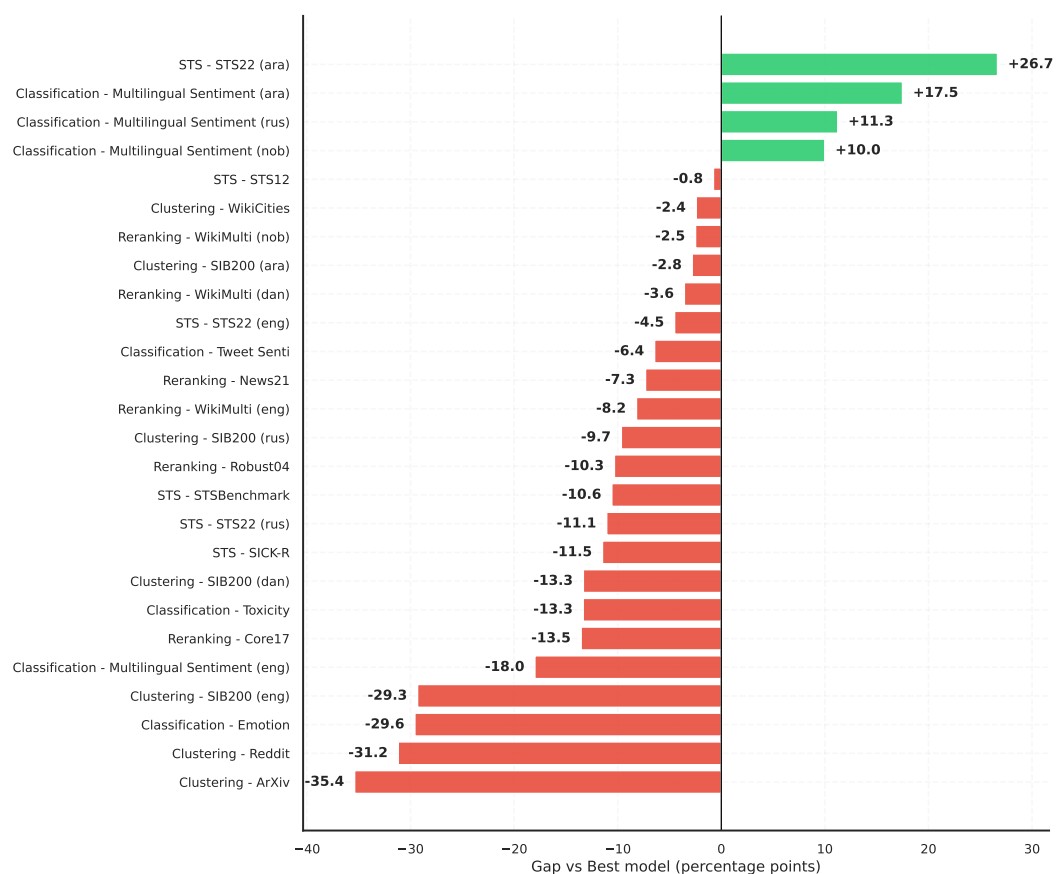

Figure 13: Human performance gaps versus best-performing models across 26 task-language pairs. Humans outperform the best models on only 4 tasks (15.4%), with largest advantages in Arabic semantic similarity and sentiment analysis. The analysis reveals systematic model advantages in technical domains (clustering, reranking) versus human advantages in culturally-informed tasks.

## H MODELS EVALUATED

Table 9 shows information about each evaluated model.

| Model | Parameters (Millions) |
|---|---|
| Alibaba-NLP/gte-Qwen2-1.5B-instruct Li et al. (2023) | 1780 |
| google/embeddinggemma-300m Vera et al. (2025) | 300 |
| intfloat/e5-mistral-7b-instruct Wang et al. (2023; 2022) | 7111 |
| intfloat/multilingual-e5-large Wang et al. (2022) | 560 |
| intfloat/multilingual-e5-base Wang et al. (2022) | 278 |
| intfloat/multilingual-e5-small Wang et al. (2022) | 118 |
| mixedbread-ai/mxbai-embed-large-v1 Lee et al. (2024); Li & Li (2023) | 335 |
| NovaSearch/jasper_en_vision_language_v1 Zhang et al. (2025a) | 1999 |
| Qwen/Qwen3-Embedding-0.6B Zhang et al. (2025b) | 596 |
| Salesforce/SFR-Embedding-Mistral (Meng et al., 2024) | 7110 |
| sentence-transformers/all-MiniLM-L6-v2 Reimers & Gurevych (2019) | 22.7 |
| sentence-transformers/all-mpnet-base-v2 Reimers & Gurevych (2019) | 109 |

Table 9: List of all evaluated models. Model sizes are in millions of parameters

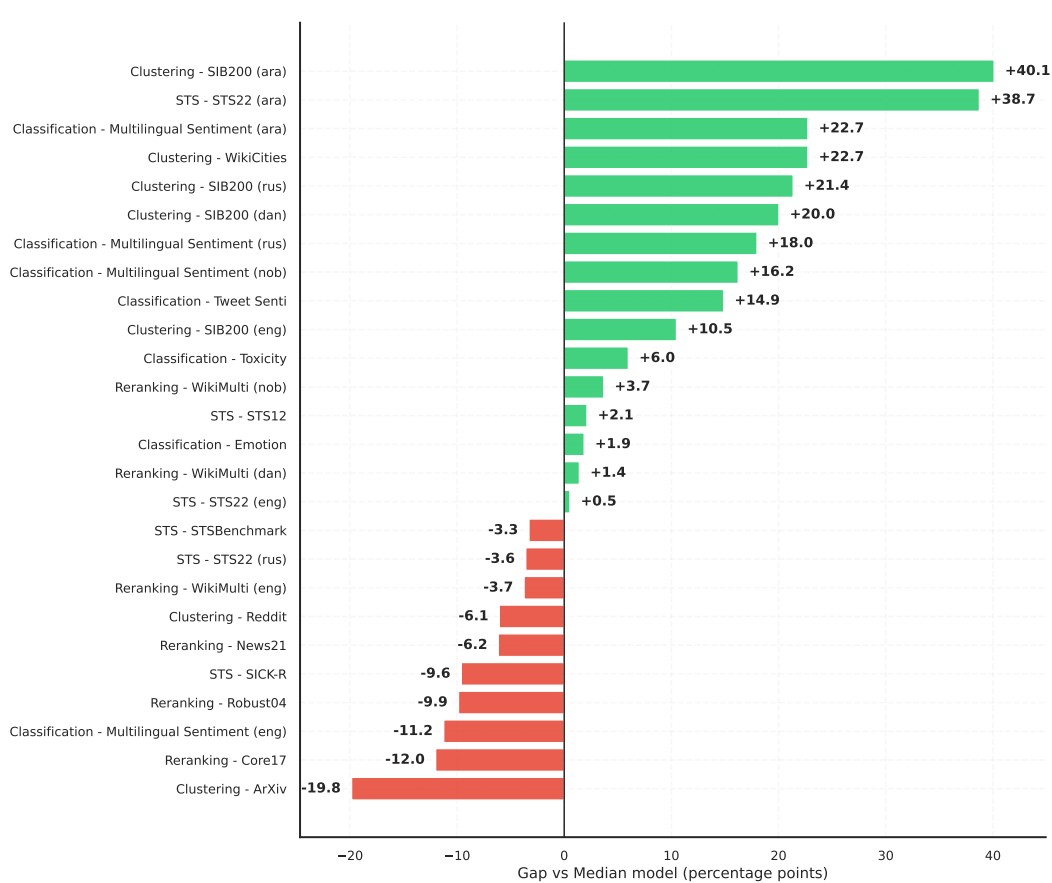

Figure 14: Human performance gaps versus median-performing models across 26 tasks by language. Humans achieve 61.5% win rate against median models, demonstrating competitive performance when compared to typical rather than best-performing models. This analysis reveals that human performance is much more competitive when compared against representative model performance rather than cherry-picked best results.

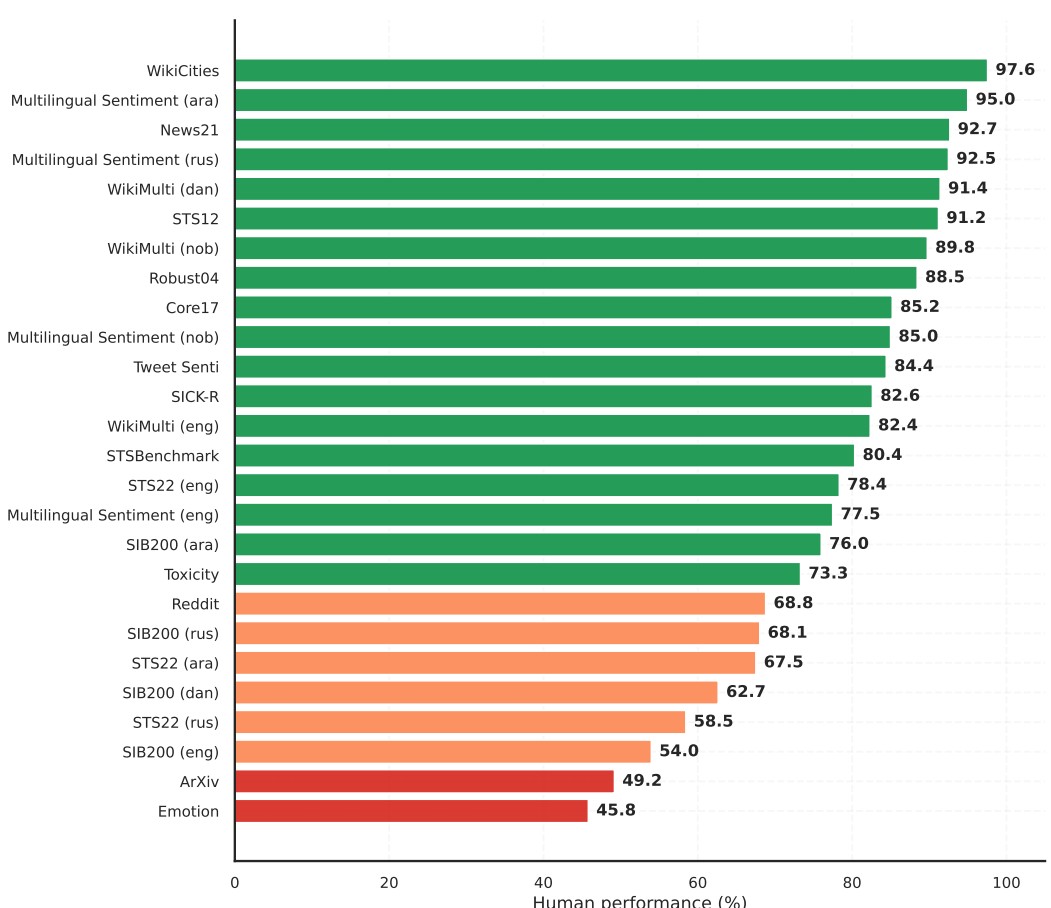

Figure 15: Task difficulty categorization based on human performance levels. The majority of tasks (69%) fall into the "easy" category (human performance $\geq 0.7$), shown in green. Only two tasks fall below $0.5$ (shown in red), both with notably low inter-annotator agreement, suggesting fundamental task ambiguity rather than limitations of human ability.

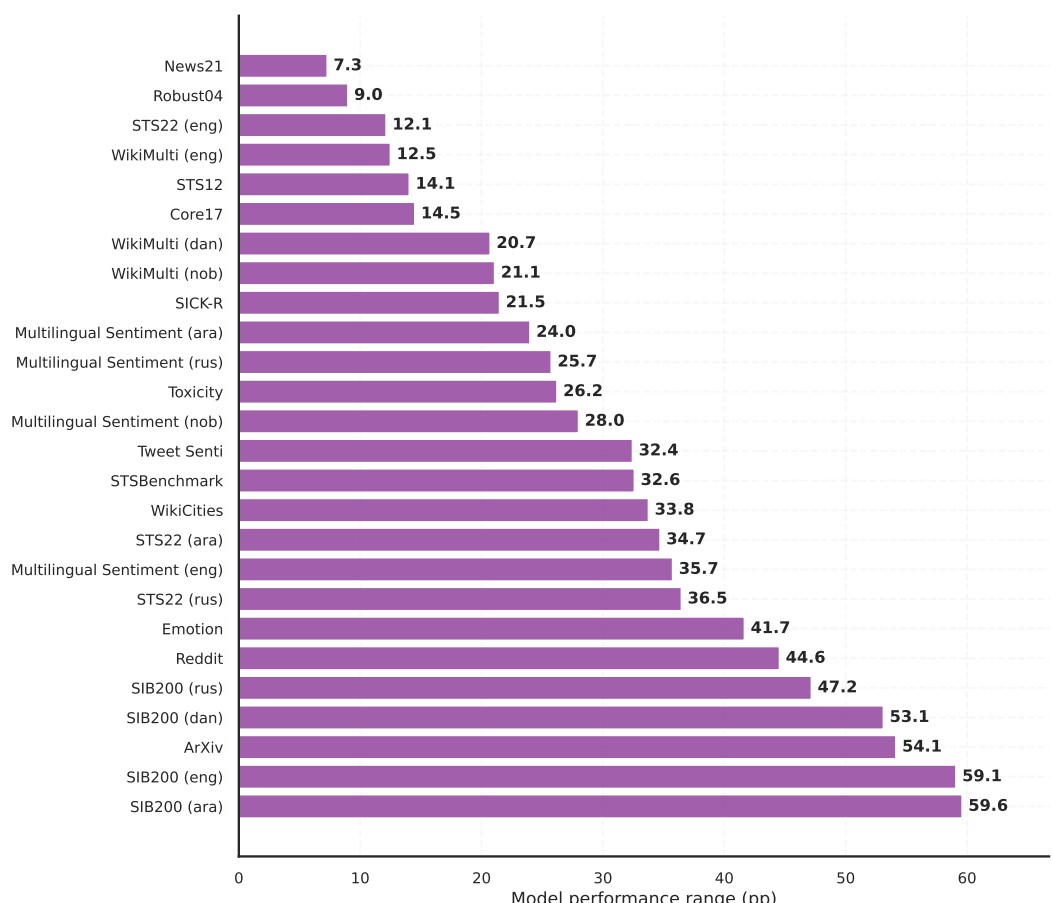

Figure 16: Model consistency analysis showing performance ranges across tasks. **Higher value indicates greater variability across models** (lower consistency). Tasks with small ranges (high consistency) often align with high human agreement, whereas tasks with large ranges (low consistency) typically correspond to tasks where humans also struggle. This pattern suggests that both human and model performance reflect underlying task quality and clarity of task specification.

# I  DETAILED LLM-AS-ANNOTATOR RESULTS

This section provides detailed LLM-as-annotator performance for each task-language pair. LLMs were not evaluated on clustering tasks due to the difficulty of eliciting cluster assignments from generative models.

| Dataset | Human | GPT-5 Full | GPT-5 Mini | GPT-4.1 Full | GPT-4.1 Mini | Gemini 2.5 Flash | Mistral Small-24B-I | Qwen3 30B | Qwen3 32B | Qwen3 Coder |
|---|---|---|---|---|---|---|---|---|---|---|
| Emotion [eng] | **45.8** | 37.5 | 33.3 | 41.7 | 29.2 | 35.4 | 43.1 | 37.5 | 38.3 | 42.1 |
| Multilingual Sentiment [ara] | **95.0** | 92.5 | 95.0 | 90.0 | 90.0 | 92.5 | 90.5 | 92.8 | 85.5 | 91.2 |
| Multilingual Sentiment [eng] | 77.5 | **100.0** | 100.0 | 97.5 | 95.0 | 97.5 | 97.2 | 99.0 | 93.0 | 96.5 |
| Multilingual Sentiment [nob] | 85.0 | **92.5** | 85.0 | 87.5 | 82.5 | 85.0 | 86.8 | 86.8 | 77.2 | 88.5 |
| Multilingual Sentiment [rus] | **92.5** | 87.5 | 85.0 | 75.0 | 85.0 | 92.5 | 64.0 | 74.2 | 75.8 | 78.0 |
| Toxic Conversations [eng] | 73.3 | 73.3 | 66.7 | 73.3 | **75.6** | 71.1 | 69.6 | 60.0 | 68.2 | 66.9 |
| Tweet Sentiment [eng] | **84.4** | 68.9 | 75.6 | 71.1 | 75.6 | 68.9 | 65.6 | 68.9 | 72.9 | 70.7 |
| **Average** | **79.1** | 78.9 | 77.2 | 76.6 | 76.1 | 77.6 | 73.8 | 74.2 | 73.0 | 76.3 |

Table 10: Detailed LLM-as-annotator results for classification tasks. **Bold** indicates best performance per row.

| Dataset | Human | GPT-5 Full | GPT-5 Mini | GPT-4.1 Full | GPT-4.1 Mini | Gemini 2.5 Flash | Mistral Small-24B-I | Qwen3 30B | Qwen3 32B | Qwen3 Coder |
|---|---|---|---|---|---|---|---|---|---|---|
| Core17 [eng] | 85.2 | 76.6 | 83.2 | 72.7 | 78.0 | 76.8 | 74.4 | **89.8** | 74.2 | 72.2 |
| News21 [eng] | **92.7** | 72.9 | 78.7 | 75.1 | 78.2 | 75.0 | 77.4 | 80.5 | 77.9 | 75.2 |
| Robust04 [eng] | **88.5** | 75.7 | 84.0 | 79.5 | 79.3 | 79.7 | 77.2 | 84.6 | 79.2 | 73.6 |
| Wikipedia [dan] | **91.4** | 84.7 | 78.3 | 85.8 | 83.6 | 82.8 | 84.0 | 76.6 | 79.8 | 79.9 |
| Wikipedia [eng] | **82.4** | 69.6 | 61.0 | 70.8 | 68.6 | 74.2 | 75.1 | 59.8 | 73.4 | 69.5 |
| Wikipedia [nob] | **89.8** | 71.5 | 68.1 | 70.6 | 75.8 | 68.6 | 79.6 | 62.6 | 64.2 | 72.3 |
| **Average** | **88.3** | 75.1 | 75.5 | 75.7 | 77.2 | 76.2 | 78.0 | 75.6 | 74.8 | 73.8 |

Table 11: Detailed LLM-as-annotator results for reranking tasks. **Bold** indicates best performance per row.

| Dataset | Human | GPT-5 Full | GPT-5 Mini | GPT-4.1 Full | GPT-4.1 Mini | Gemini 2.5 Flash | Mistral Small-24B-I | Qwen3 30B | Qwen3 32B | Qwen3 Coder |
|---|---|---|---|---|---|---|---|---|---|---|
| SICK-R [eng] | **82.6** | 68.5 | 67.5 | 66.9 | 72.9 | 59.0 | 66.7 | 59.4 | 57.3 | 68.8 |
| STS12 [eng] | **91.2** | 83.2 | 82.7 | 87.3 | 87.8 | 83.8 | 83.9 | 83.8 | 81.7 | 84.5 |
| STS22 [ara] | **67.5** | 57.1 | 45.5 | 56.9 | 55.8 | 43.1 | 56.9 | 35.3 | 50.4 | 47.9 |
| STS22 [eng] | 78.4 | 67.8 | 80.3 | **81.6** | 79.7 | 81.3 | 78.2 | 74.1 | 76.2 | 76.4 |
| STS22 [rus] | 58.7 | 71.9 | 51.3 | 67.2 | 66.6 | 64.9 | **78.6** | 65.9 | 62.4 | 67.0 |
| STSBenchmark [eng] | 80.4 | 89.3 | 86.7 | **89.7** | 86.3 | 83.8 | 85.4 | 84.3 | 84.0 | 82.9 |
| **Average** | **76.5** | 73.0 | 69.0 | 74.9 | 74.9 | 69.3 | 75.0 | 67.1 | 68.6 | 71.3 |

Table 12: Detailed LLM-as-annotator results for sts tasks. **Bold** indicates best performance per row.

## I.1  HUMAN-LLM TASK DIFFICULTY CORRELATION

To assess whether humans and LLMs face similar challenges, we computed Spearman rank correlations between human and LLM performance across the 19 task-language pairs (excluding clustering). A positive correlation indicates that tasks where humans perform well also tend to be tasks where LLMs perform well.

The overall correlation is moderate and statistically significant ($\rho = 0.52$, $p = 0.023$), suggesting partially shared difficulty patterns.

| LLM Model | Spearman $\rho$ | $p$-value |
|---|---|---|
| GPT-5 Full | 0.47 | 0.042 |
| GPT-5 Mini | 0.52 | 0.023 |
| GPT-4.1 Full | 0.43 | 0.066 |
| GPT-4.1 Mini | 0.57 | 0.012 |
| Gemini 2.5 Flash | 0.50 | 0.029 |
| Mistral Small | 0.30 | 0.215 |
| Qwen3-30B | 0.50 | 0.031 |
| Qwen3-32B | 0.52 | 0.022 |
| Qwen3-Coder | 0.57 | 0.011 |
| **Average (all LLMs)** | **0.52** | **0.023** |

Table 13: Spearman rank correlation between human and LLM performance across 19 task-language pairs (clustering excluded). A positive $\rho$ indicates that tasks where humans score high also tend to be tasks where LLMs score high. Values with $p < 0.05$ indicate statistically significant correlations. The moderate positive correlations suggest partially shared task difficulty patterns between humans and LLMs.

## J   LLM USAGE STATEMENT

Large language models were used to assist with formatting, citation integration, and writing polish during the preparation of this manuscript. Specifically, we used LLMs for:

- Formatting assistance for LaTeX tables and mathematical notation
- Integration and standardization of citation formats
- Minor writing improvements for clarity and flow
- Code documentation and data processing script organization

All substantive content, including research design, data analysis, interpretation of results, and scientific conclusions, was developed entirely by the authors. The core contributions, methodology, and findings presented in this work are the original intellectual contribution of the research team. LLM assistance was limited to technical formatting and presentation improvements that did not influence the scientific content or conclusions of the study.