# OpenReview forum: "HUME: Measuring the Human-Model Performance Gap in Text Embedding Tasks"
_ICLR.cc/2026/Conference — ICLR 2026 Poster_

### Official Review · Reviewer_N1ED · 2025-10-18

**Soundness:** 2
**Presentation:** 3
**Contribution:** 2
**Rating:** 4
**Confidence:** 4

**Summary:**

This paper introduces HUME, a framework for measuring human performance on text embedding tasks from MTEB. The authors evaluate human annotators across 16 datasets spanning four task categories: reranking, classification, clustering, and semantic textual similarity (STS).

**Strengths:**

The selection of 16 diverse datasets across multiple languages, domains, and task types is well-motivated and thorough. The inclusion of both high-resource and low-resource languages adds an important cross-lingual perspective.

**Weaknesses:**

While the paper addresses an interesting question, I have concerns:
1. **Necessity and Scope of the Study**
	- **Limited sample sizes**: With only 20-50 instances per task, can we really draw reliable conclusions about human performance on these benchmarks?
	- **Missing task categories**: The framework only covers 4 of MTEB's task types (reranking, classification, clustering, STS). What about retrieval, which is a very important task nowadays?
2. **Methodological Concerns**
	- **Statistical significance**: The paper presents many performance comparisons but doesn't include significance testing. With small sample sizes, how confident can we be that observed differences are meaningful?

**Questions:**

**Unclear Motivation for Human-Model Comparison**

I'm struggling to understand the fundamental premise of this paper. Why should we compare embedding models to human performance at all?

This isn't like evaluating generative models, where we compare LLMs to humans because we want human-level intelligence and knowledge. Embedding models serve a completely different purpose. We never use humans to generate embeddings or perform large-scale semantic search in practice. The goal is optimizing downstream task performance, not replicating human judgment. So what does human performance actually tell us about embedding quality?

The paper repeatedly interprets superior model performance as suspicious, suggesting models are "exploiting patterns rather than achieving true semantic understanding." But why can't models just be legitimately better at these tasks? Machine learning is designed to find patterns beyond human perception. When a model outperforms humans at emotion classification, maybe it's actually better at detecting emotional patterns consistently, rather than exploiting flawed data.

---

> ### Author Response · Authors · 2025-11-18
>
> We thank the reviewer for the detailed feedback.
>
> > While the paper addresses an interesting question, I have concerns:
> > Necessity and Scope of the Study
> > Limited sample sizes: With only 20-50 instances per task, can we really draw reliable conclusions about human performance on >these benchmarks?
> >Missing task categories: The framework only covers 4 of MTEB's task types (reranking, classification, clustering, STS). What about >retrieval, which is a very important task nowadays?
> >Methodological Concerns
> >Statistical significance: The paper presents many performance comparisons but doesn't include significance testing. With small >sample sizes, how confident can we be that observed differences are meaningful?
>
> On limited sample sizes & statistical significance: This is a valid critique. As detailed in General Comment #3, we will add 95% confidence intervals and statistical significance testing in the revision to address concerns about the robustness of our findings with the current sample sizes.
> On missing retrieval tasks: Thank you for raising this point. As noted in General Comment #2, we used reranking as a human-evaluable proxy for retrieval, as direct human evaluation of large-scale retrieval would require annotators to process thousands of candidate documents per query.
>
> > Unclear Motivation for Human-Model Comparison
>
> This is a critical point that we address in General Comment #1, but we want to clarify further here, as there may be a misunderstanding of our core premise.
>
> Our goal is benchmark calibration and quality assessment, not setting a performance ceiling. We agree completely that embedding models can and should exceed human performance on well-defined tasks: this is the entire point of machine learning. We are not arguing that human performance is an upper bound.
>
> However, the key question is: how do we know when a model is truly "better" versus simply overfitting to flawed labels? This is where human evaluation becomes essential as a diagnostic tool.
>
> Our central finding directly addresses this: models frequently achieve "superhuman" performance specifically on datasets where humans themselves cannot agree (low inter-annotator agreement/IAA). This pattern is the critical evidence. When humans show high agreement on a task (high IAA), we would interpret superior model performance exactly as you suggest: as genuine capability. But when humans fundamentally disagree on what the "correct" answer is, the ground truth labels themselves are ambiguous or subjective.
>
> In these low-IAA cases, it is impossible to determine whether the model has superior semantic understanding or is simply better at predicting the arbitrary patterns in noisy labels. A model that achieves 95% accuracy by perfectly memorizing ambiguous labels is not demonstrating semantic understanding: it's exploiting dataset artifacts.
>
> HUME provides the diagnostic tool to distinguish between these scenarios. Without measuring human agreement, we cannot tell if a benchmark score of 95% means "the model has excellent semantic understanding" or "the model has overfit to a flawed dataset." Given the current quality issues we identified in MTEB, we simply cannot interpret many "superhuman" results as evidence of superior capability.
>
> In the revision, we will add more concrete examples in the main text showing cases where models achieve high accuracy on items with low human agreement, making this distinction clear and explicit.

---

> > ### Comment · Reviewer_N1ED · 2025-11-28
> >
> > Thanks for the work, but I still have some concerns about the motivation.
> >
> > First, measuring inter-annotator agreement isn't new; it's been standard practice in NLP for decades. You're essentially auditing MTEB datasets for IAA and finding quality issues, which isn't particularly surprising. Second, putting humans on the same leaderboard as embedding models is misleading since they're fundamentally doing different things. Finally, I am still wondering about the practical value here. Practitioners will likely continue using MTEB and optimizing for downstream performance regardless of these findings.

---

> ### Author Response · Authors · 2025-12-01
>
> We thank the reviewer for their continued engagement. We understand the concern that IAA audits are standard practice and that humans and models process text differently. However, we believe there is a critical distinction between the method (which is standard) and the application (auditing the specific benchmarks that currently drive the field), which has immense practical implications.
>
> >First, measuring inter-annotator agreement isn't new; it's been standard practice in NLP for decades. You're essentially auditing MTEB datasets for IAA and finding quality issues, which isn't particularly surprising
>
> While checking Inter-Annotator Agreement (IAA) is indeed a standard method in NLP, it has effectively been absent from the application of the field's most influential leaderboard (MTEB). The fact that this is "standard practice" makes its absence in embedding benchmarks even more critical to rectify. We are essentially retrofitting standard scientific rigor to a benchmark that thousands of practitioners rely on blindly. If the findings (that models exploit artifacts on low-IAA datasets) seem unsurprising in hindsight, they are nonetheless critical because SOTA models are currently claiming "victory" on these very datasets. The community is mistaking fitting to annotation noise for semantic progress; our work provides the empirical evidence required to correct this.
>
> >Second, putting humans on the same leaderboard as embedding models is misleading since they're fundamentally doing different things.
>
> We agree with the reviewer that humans and models process text differently on a mechanistic level. However, the fundamental objective of a text embedding is to act as a proxy for human semantic perception. Human consensus therefore represents the "gold standard" upper bound for validity. If a model achieves a correlation score higher than the agreement rate of the humans (who defined the ground truth), the model is mathematically overfitting to artifacts rather than capturing "super-human" semantic meaning. Placing humans on the leaderboard is the only way to visualize where models have crossed the threshold from "useful optimization" to "overfitting noise."
>
> > Finally, I am still wondering about the practical value here. Practitioners will likely continue using MTEB and optimizing for downstream performance regardless of these findings.
>
> Practitioners care most about reliability. If a model achieves a high MTEB score but fails in production because the benchmark was measuring annotation artifacts rather than semantic meaning, the leaderboard loses its utility. Our framework provides the "validity check" required to restore trust in the scores. To ensure this work translates into immediate impact, we are actively coordinating with the MTEB maintainers to operationalize these findings. We note that similar community feedback was instrumental in the transition from MTEB(eng, v1) (Muennighoff et al., 2022) to MTEB(eng, v2). By directly influencing the composition of future benchmarks, this work ensures that the field optimizes for genuine semantic capability rather than dataset noise.
>
>
> We will update our discussion in the final revision of the paper to highlight the concrete steps to improve future benchmarks based on these findings and further refine motivation.

---

### Official Review · Reviewer_zWFZ · 2025-10-21

**Soundness:** 2
**Presentation:** 3
**Contribution:** 3
**Rating:** 4
**Confidence:** 4

**Summary:**

The paper introduces HUME, a framework for measuring human performance on text embedding tasks and comparing it directly to state-of-the-art embedding models. It evaluates human annotators across 16 MTEB datasets spanning classification, clustering, reranking, and semantic textual similarity in multiple languages. Results show that human performance is competitive but not dominant, often ranking around the upper-middle of model performance, with notable advantages in certain multilingual tasks. The authors analyze task difficulty, dataset quality, and human agreement, highlighting that low human performance often reflects dataset ambiguity rather than human limitations. The paper concludes with implications for benchmark design, cultural and linguistic gaps in current models, and recommendations for more reliable evaluation practices.

**Strengths:**

1. The paper tackles a timely and important problem by grounding text embedding evaluation in human performance. The motivation is well articulated, and the direction is practical and relevant for improving how benchmark results are interpreted.

2. The experimental design is generally solid, with clear task selection and consistent evaluation protocols. While some aspects could be expanded (see weaknesses), it establishes a strong foundation for systematic human–model comparison across languages and task types.

3. The discussion and implications are well developed, offering clear insights into dataset quality, evaluation reliability, and multilingual gaps. The proposed future suggestions are concrete and meaningful, adding to the significance and forward-looking value of the work.

**Weaknesses:**

1. The study’s reliance on only two annotators for most tasks (and, in some multilingual settings, just one) severely limits the robustness and representativeness of the claimed “human” performance. With such a small pool, the results risk reflecting individual annotator biases rather than general human ability, making the observed human–model gap less reliable as an empirical reference point. In addition, the number of annotated examples per dataset—ranging only from 30 to 50 items—is far too small relative to the scale of the original benchmarks. This raises concerns about statistical power and generalizability: subtle effects may be missed, while task-specific variability could be overstated or underrepresented in the final conclusions.

2. Although the paper identifies several dataset quality issues, it does not sufficiently analyze their underlying causes from the perspective of model training. In particular, there is no supporting evidence or exploration of factors such as training data distributions, domain coverage, or cultural mismatches that might explain the observed human–model performance differences. This lack of analytical depth weakens the explanatory power of the findings, especially in multilingual settings where humans outperform models in some tasks but not others.

3. The paper directly compares human annotators and embedding models but lacks a key experiment: whether large language models can follow the same annotation protocol. Evaluating LLMs under the same instructions would not only provide a useful reference baseline but also offer practical value by reducing the cost of future human evaluations. This experiment could significantly strengthen the paper’s claims regarding the human–model gap.

4. While the paper covers classification, clustering, STS, and reranking tasks, it notably omits retrieval tasks, which are arguably the most central and widely used application of text embeddings in real-world systems. Excluding retrieval significantly limits the practical relevance and completeness of the study’s conclusions.

5. The paper provides insufficient clarity on whether the human labeling instructions were fully aligned with the original dataset annotation schemes. Any divergence in labeling guidelines could introduce inconsistencies and confound the comparison between human annotators and model performance, raising questions about the validity of the measured human–model gap.

**Questions:**

1. (W1) Could the authors provide justification for using such a small number of annotators and annotated samples? Do they have any evidence or pilot studies suggesting that this limited pool and sample size are sufficient to produce stable and representative estimates of “human” performance?

2. (W2) Can the authors provide a more detailed analysis of the underlying factors contributing to the observed human–model performance differences, such as training data coverage, domain bias, or cultural and linguistic variability, especially in multilingual tasks?

3. (W3) Would it be feasible to evaluate LLMs by instructing them to follow the same annotation protocol as human annotators? If so, how might such an experiment serve as a complementary or cost-efficient reference point in future work?

4. (W4) Why were retrieval tasks excluded from the study, and how might including them—given their centrality to text embedding applications—affect the overall findings and implications?

5. (W5) How closely were the human labeling instructions matched to the original dataset annotation schemes, and could any inconsistencies between the two affect the validity of the human–model comparisons?

---

> ### Author Response · Authors · 2025-11-18
>
> We thank the reviewer for the detailed feedback and constructive suggestions.
>
> > (W1) Could the authors provide justification for using such a small number of annotators and annotated samples? Do they have any evidence or pilot studies suggesting that this limited pool and sample size are sufficient to produce stable and representative estimates of “human” performance?
>
> This is a key concern, which we address in General Comment #3. We will add confidence intervals and significance testing to bolster the statistical robustness of our "breadth-first" findings. We would also like to emphasize that the diversity of tasks and languages (16 tasks across 5 languages) was a core design goal, enabling us to identify cross-task patterns that would not be visible in a narrow, deep study.
>
> > (W2) Can the authors provide a more detailed analysis of the underlying factors contributing to the observed human–model performance differences, such as training data coverage, domain bias, or cultural and linguistic variability, especially in multilingual tasks?
>
> This is an excellent point. You are correct that we do not deeply analyze the root causes of these differences (e.g., training data distributions, cultural biases). Unfortunately, for many models, this information is not publicly available. However, we will add a section to our discussion explicitly acknowledging this limitation and framing it as a critical direction for future work. Importantly, we will note that our HUME framework provides the necessary diagnostic tool to enable such investigations. By establishing human baselines, we can now identify *where* models diverge from human performance, which is the first step toward understanding *why*.
>
>
> > (W3) Would it be feasible to evaluate LLMs by instructing them to follow the same annotation protocol as human annotators? If so, how might such an experiment serve as a complementary or cost-efficient reference point in future work?
>
> This is an excellent and constructive suggestion. We fully agree that this is a promising approach, comparing LLM-as-annotator performance would provide valuable context and potential cost savings for future evaluations. We will conduct a preliminary experiment evaluating a top frontier LLM on a subset of our tasks using identical instructions. If time permits, we will include these results in our revised paper as a new and valuable ablation; otherwise, we will add this as a concrete direction for future work in our discussion, noting its potential value for scaling human evaluation.
>
>
> > (W4) Why were retrieval tasks excluded from the study, and how might including them—given their centrality to text embedding applications—affect the overall findings and implications?
>
> Thank you for raising this point. As noted in General Comment #2, we used reranking as a human-evaluable proxy for retrieval, as direct human evaluation of large-scale retrieval would require annotators to process thousands of candidate documents per query.
>
> > (W5) How closely were the human labeling instructions matched to the original dataset annotation schemes, and could any inconsistencies between the two affect the validity of the human–model comparisons?
>
> This is a valid concern. Our human labeling instructions were meticulously designed to match the task definitions exactly (e.g., identical label sets for classification, same 1-5 scale for STS) to ensure valid comparisons. However, we acknowledge that formal, detailed annotation protocols are not publicly available for many MTEB datasets, which limits our ability to verify perfect alignment. In the revision, we will explicitly state our instruction design process in the methodology section and acknowledge this potential source of variance. This limitation further underscores our finding that many MTEB datasets may have underlying quality issues.

---

> > ### Comment · Reviewer_zWFZ · 2025-11-19
> >
> > Thank you for your reply. I have read both the general response and the specific response to my review. I agree that there is no perfect method, and I did not find any major misunderstanding of my comments. I appreciate the motivation behind your work and the core idea you are pursuing, though I still have some concerns regarding the current version.
> >
> > That said, I encourage you not to be discouraged by the borderline scores. Instead, I strongly recommend taking some time to dive deeper into the issues raised and further improve the paper. I believe the work has potential, and a more refined version could make a stronger impact.
> >
> > I look forward to seeing your revision, along with whatever experiments are feasible within the short discussion period!

---

### Official Review · Reviewer_RywH · 2025-10-29

**Soundness:** 3
**Presentation:** 2
**Contribution:** 3
**Rating:** 4
**Confidence:** 4

**Summary:**

This paper introduces HUME, a framework for measuring human-level performance on text embedding benchmarks. They compared human performance with 13 embedding models on 4 major categories of tasks (classification, clustering, semantic textual similarity, and reranking) in MTEB and analyzed the consistency of human annotators as the entry point, putting forward suggestions for the development of future evaluation benchmarks.
The main contributions include: (1) Proposing a framework for evaluating human embedding capabilities, which is currently relatively lacking; (2) The assumption that tasks with low human consistency may reflect the ambiguity of data or definitions rather than the true "superhuman" model capabilities offers a new guidance for the future development of benchmarks.

**Strengths:**

Supper cool idea!
1. The paper is original in reframing benchmark evaluation: a human evaluation framework that quantifies human-level performance across text-embedding tasks and relates it to model results.
2. The study is of good methodological quality, covering sixteen datasets and four task categories with transparent protocols and clear reporting.
3. The paper’s significance lies in revealing that many “superhuman” model claims arise in tasks with low human agreement, highlighting the need for benchmark reform and more interpretable evaluation standards.

**Weaknesses:**

The small and homogeneous annotator pool (all male, limited in number, with single annotators for some languages) limits generalizability and the evaluation sets are small and lack confidence interval reporting, reducing statistical robustness.

There is a lack of a more direct and powerful analysis of the performance attribution of the superhuman model (directly attributing low human consistency to data/task design and quality issues is not rigorous enough).

The retrieval is the most widely used embeddinng application, which is not included in this work. I'm very curious and excited about how well this framework could guide the retriver evaluation.

Additionally, the proposed “agreement-weighted evaluation” idea is conceptually interesting but underdeveloped.

**Questions:**

How do you ensure that the low agreement among your annotators truly reflects data ambiguity, rather than annotation fatigue or lack of clear instructions? It is possible to supplement the analysis of model bad cases and whether the model as a whole leans towards certain labels.

Could you share a concrete formula for your proposed “agreement-weighted evaluation”? It is suggested that several feasible agreement-based weighted evaluation methods be proposed.

**Details Of Ethics Concerns:**

Is there a description of the annotator’s compensation, protection and psychological safety, etc. details?

---

> ### Author Response · Authors · 2025-11-18
>
> We thank the reviewer for the enthusiastic feedback, particularly for finding our core idea "super cool" and for the detailed engagement with our methodology.
>
> > The small and homogeneous annotator pool (all male, limited in number, with single annotators for some languages) limits generalizability and the evaluation sets are small and lack confidence interval reporting, reducing statistical robustness.
>
> > Flag For Ethics Review
>
> Thank you for raising these critical points regarding ethical practice and methodological scope. We acknowledge that clarity on these fronts is essential.
>
> Ethical and Diversity Considerations: We apologize for the omission of ethical details. We will add a dedicated "Ethical Considerations" section in the final revision detailing that annotators were co-authors who consented to the study. While our initial annotator pool had limitations in size and demographic homogeneity, we stress that our recruitment was designed for crucial linguistic and cultural diversity across five distinct languages (English, Arabic, Russian, Norwegian Bokmål, Danish). We consider this linguistic breadth a primary, non-trivial focus of our study.
>
> Statistical Robustness: Regarding statistical robustness, we agree that confidence metrics are necessary. As detailed in General Comment #3, we will add 95% confidence intervals to our main results. Furthermore, we will move our comprehensive Inter-Annotator Agreement (IAA) metrics (currently in Appendix D) to the main paper to directly support our central claims and enhance transparency.
>
> > There is a lack of a more direct and powerful analysis of the performance attribution of the superhuman model (directly attributing low human consistency to data/task design and quality issues is not rigorous enough).
>
> > How do you ensure that the low agreement among your annotators truly reflects data ambiguity, rather than annotation fatigue or lack of clear instructions? It is possible to supplement the analysis of model bad cases and whether the model as a whole leans towards certain labels.
>
> This is a key methodological point. We ensured clear instructions and manageable batch sizes to minimize fatigue. More importantly, our qualitative analysis (currently in Appendix D) indicates that disagreements stem primarily from inherent semantic ambiguity in the "ground truth" (e.g., the subjective nature of "emotion" or "intent") rather than random error associated with fatigue.
> In the revision, we will strengthen this analysis by: (1) moving key examples from Appendix D to the main text, and (2) adding concrete "bad cases" where models are technically "correct" (matching the ground truth label) but the label itself is clearly ambiguous, demonstrating that models are exploiting these ambiguous labels rather than achieving true semantic understanding. While a comprehensive quantification of model label bias is outside our current scope, we will include qualitative observations about any apparent label preferences in the bad case analysis.
>
> > The retrieval is the most widely used embeddinng application, which is not included in this work. I'm very curious and excited about how well this framework could guide the retriver evaluation.
>
> Thank you for raising this point. As noted in General Comment #2, we used reranking as a human-evaluable proxy for retrieval.
>
> > Additionally, the proposed “agreement-weighted evaluation” idea is conceptually interesting but underdeveloped.
>
> > Could you share a concrete formula for your proposed “agreement-weighted evaluation”?
>
> Thank you for highlighting this. We acknowledge that this concept was underdeveloped in our submission. To clarify: we are not proposing a novel agreement-weighted formula, but rather advocating that future embedding benchmarks adopt established methodologies that respect label disagreement.
> Multiple existing works have already developed rigorous frameworks for evaluation and training while accounting for annotator disagreement, notably the body of work by Barbara Plank and colleagues, which includes:
>
> Learning part-of-speech taggers with inter-annotator agreement loss (Plank et al.)
>
> We Need to Consider Disagreement in Evaluation (Basile et al.)
>
> Beyond Black & White: Leveraging Annotator Disagreement via Soft-Label Multi-Task Learning (Fornaciari et al.)
>
> These works provide concrete methods such as soft-label learning and inter-annotator agreement loss that could be directly applied to embedding evaluation. In our revision, we will explicitly cite these frameworks and recommend their adoption for handling the label ambiguity we identified in MTEB, rather than continuing to rely solely on hard "ground truth" accuracy metrics.
> We agree that this connection was not clear in the current write-up and will remedy this in the final revision.

---

> > ### Comment · Reviewer_RywH · 2025-11-26
> >
> > Thank authors for the detailed response.
> >
> > My misunderstanding about the “agreement-weighted evaluation” has been resolved. I encourage the authors to further explore this aspect to strengthen the paper.
> >
> > I agree that using reranking as a proxy for retrieval is reasonable; indeed, on MTEB, reranking is kind of a retrieval (by embedding-based retreival) task, with a much smaller candidate pool.
> >
> > I encourage authors to further refine and strengthen this work, with the aim of achieving higher review scores in future submissions.

---

### Official Review · Reviewer_FCbW · 2025-11-01

**Soundness:** 2
**Presentation:** 2
**Contribution:** 2
**Rating:** 4
**Confidence:** 3

**Summary:**

The paper introduces HUMA, a human performance evaluation on the MTEB benchmark. Results show that humans achieve an average score of 77.6%, compared to 80.1% for the best embedding model. Further analysis reveals that text embedding models underperform humans on low-resource languages, and that some datasets exhibit low quality or label ambiguity.

**Strengths:**

(1) The paper presents a comprehensive human performance evaluation of MTEB, covering reranking, classification, clustering, and STS datasets.

(2) Based on the human performance results and inter-annotator agreement analysis, the authors identify several problematic datasets containing labeling ambiguities (e.g., the emotion classification dataset), which may lead to unrealistic model evaluations..

**Weaknesses:**

(1) The motivation of conducting a comprehensive human performance evaluation of MTEB is not very convincing to me. The two possible reasons I can get are (1) estimating an uppper bound of dataset performance, similar to how human performance on GLUE was previously used as a target to achieve. But this is not the case here since the paper shows that current models already surpass human performance; and (2) as the paper points out, identifying problematic datasets where humans perform poorly or where inter-annotator agreement is low. However, this second motivation seems somewhat weak to me.

(2) The retrieval task, which is widely used in real-world applications, is not included in the human performance evaluation.

(3) Most tasks includes only two annotations, and for each task there are only 20-50 instances annotated. This raise concerns on the reliablity of the evaluation.

(4) Comparing the performance of native low-resource language speakers with text embedding models primarily trained on English data may not be a fair comparison.

**Questions:**

(1) In L99-102, the paper states that "Human evaluation is well established in NLP, especially for generative tasks like machine translation, summarization, and dialogue.  In contrast, embedding-based tasks have relied almost exclusively on automated metrics, with little attention to human baselines.". This motivation is not convincing to me. For these generation tasks, automatic evaluation metrics are often unreliable, which requires human evaluation. However, for embedding-based tasks that have well-defined ground truth labels, human evaluation may not be that necessary.

---

> ### Author Response · Authors · 2025-11-18
>
> Thank you for the clear summary and for noting the comprehensive nature of our evaluation.
>
>
> > (1) The motivation of conducting a comprehensive human performance evaluation of MTEB is not very convincing to me. The two possible reasons I can get are (1) estimating an uppper bound of dataset performance, similar to how human performance on GLUE was previously used as a target to achieve. But this is not the case here since the paper shows that current models already surpass human performance; and (2) as the paper points out, identifying problematic datasets where humans perform poorly or where inter-annotator agreement is low. However, this second motivation seems somewhat weak to me.
>
> > (1) In L99-102, the paper states that "Human evaluation is well established in NLP, especially for generative tasks like machine translation, summarization, and dialogue. In contrast, embedding-based tasks have relied almost exclusively on automated metrics, with little attention to human baselines.". This motivation is not convincing to me. For these generation tasks, automatic evaluation metrics are often unreliable, which requires human evaluation. However, for embedding-based tasks that have well-defined ground truth labels, human evaluation may not be that necessary.
>
> This is the most important point, which we discussed in detail in General Comment #1. Our motivation is primarily diagnostic. We find that the "ground-truth" labels are often ambiguous, as evidenced by low human agreement. Human evaluation is necessary to diagnose the quality of the ground truth itself and to interpret model scores meaningfully.  We will ensure this diagnostic framing is made explicitly clear in the final revision.
>
> > (2) The retrieval task, which is widely used in real-world applications, is not included in the human performance evaluation.
>
> We address this in General Comment #2. We utilize reranking as a necessary and human-evaluable proxy for retrieval, and we will clarify this methodological choice in the paper.
>
> > (3) Most tasks includes only two annotations, and for each task there are only 20-50 instances annotated. This raise concerns on the reliablity of the evaluation.
>
> We address this in General Comment #3. To bolster our "breadth-first" results, we will add confidence intervals and significance testing to the revision.
>
> > (4) Comparing the performance of native low-resource language speakers with text embedding models primarily trained on English data may not be a fair comparison.
>
> This is an excellent point, and our framing may have been unclear. This comparison was intended to highlight this exact gap. The fact that English-centric models underperform native speakers is a key finding, demonstrating a clear shortcoming and linguistic/cultural gap in current SOTA models. We will clarify this framing in our discussion.

---

> > ### Comment · Reviewer_FCbW · 2025-11-27
> >
> > Thank you for the detailed response.
> >
> > Regarding point (1), as I noted in my original review "However, this second motivation (diagnostic) seems somewhat weak to me.", to me conducting human evaluation on MTEB datasets to identify problemetic datasets seems a somewhat weak contribution. And the findings that tasks with vague lables can be probelimetic is also not fully interesing to me. I was wondering if the author could clarify the motivation more?
> >
> > Regarding point (3), since ICLR allows PDF updates, I was wondering if the authors could directly update the PDF to inclde the confidence intervals and significance testing results?
> >
> > My concerns regarding Weaknesses (2) and (4) have been effectively addressed.

---

> ### Author Response · Authors · 2025-12-01
>
> We thank the reviewer for the continued engagement and for acknowledging that their concerns regarding weaknesses (2) and (4) have been addressed.
>
> > Regarding point (1), as I noted in my original review "However, this second motivation (diagnostic) seems somewhat weak to me.", to me conducting human evaluation on MTEB datasets to identify problemetic datasets seems a somewhat weak contribution. And the findings that tasks with vague lables can be probelimetic is also not fully interesing to me. I was wondering if the author could clarify the motivation more?
>
> We appreciate the opportunity to clarify why identifying problematic datasets via human evaluation is a critical, rather than weak, contribution. Our motivation extends beyond simply labeling datasets as "vague."
>
> Benchmarks like MTEB currently steer the research and development of the field. When these benchmarks contain undetected noise or ambiguity, the field collectively makes subpar decisions regarding architecture selection and deployment. Without the "audit" provided by HUME, researchers risk "blind optimization": expending significant R&D resources to improve scores on datasets where the "ground truth" is actually annotation artifact.
>
> The contribution here is the empirical evidence that current SOTA models are likely overfitting to noise in specific tasks (e.g., Emotion Classification, where models far exceed human agreement). By identifying these distinct failure modes, we prevent the community from mistaking "fitting to noise" for "semantic progress." This directly impacts the efficiency of new architecture development and the reliability of models selected for real-world application.
>
> To demonstrate that this diagnostic motivation translates into immediate impact, we are actively coordinating with the MTEB maintainers to operationalize these findings. We note that similar community feedback was instrumental in the transition from MTEB(eng, v1) (Muennighoff et al., 2022) to MTEB(eng, v2). Thus, our work is not just observing issues; it is actively shaping the standard by which future models will be evaluated.
>
> > Regarding point (3), since ICLR allows PDF updates, I was wondering if the authors could directly update the PDF to inclde the confidence intervals and significance testing results?
>
> We have added 95% confidence intervals to our result tables and included the statistical significance testing results to ensure the robustness of our claims, as requested. We will be pushing the final PDF manuscript with the updated changes shortly.
>
>
> We will be updating our discussion in the final revision of the paper to explicitly articulate this argument regarding the high cost of unreliable benchmarks and further refine motivation.

---

### Author Response · Authors · 2025-11-18

We sincerely thank all four reviewers (FCbW, RywH, zWFZ, N1ED) for their time and for providing detailed, insightful, and constructive feedback on our submission. We are encouraged that reviewers found our core idea "supper cool" (RywH) and the problem "timely and important" (zWFZ).
The feedback has provided a clear and valuable path for strengthening our work. The most critical feedback converged on three themes, which we address first in a general response, followed by specific replies to each reviewer.


1. On the Core Motivation: Calibration, Not a Ceiling (Response to FCbW, N1ED)

Several reviewers (FCbW, N1ED) correctly questioned the motivation for comparing human and model performance, arguing that "ground-truth" labels already exist and that "human-level" performance is not necessarily the goal for embeddings.
Our goal is benchmark calibration and diagnostics, not setting a performance "ceiling." We are not using human performance as an upper bound, but as a tool to interpret benchmark scores and diagnose dataset quality. Our work questions the "ground truth" itself. As our results and low inter-annotator agreement (IAA) show, many labels in MTEB are ambiguous, subjective, or of poor quality.
The key finding is that when models achieve "superhuman" performance, it often occurs on datasets with low IAA. This suggests that models are not achieving true "superhuman" semantic understanding, but are instead exploiting label ambiguity or overfitting to artifacts in the benchmark's "ground truth." Thus, HUME provides an essential diagnostic. It allows us to differentiate between (a) models truly excelling at a well-defined task and (b) models simply overfitting to a flawed dataset. In the final revision, we will add concrete examples to the main text where the model is 'correct' (i.e., matches the 'ground truth') but the label itself is clearly ambiguous, underscoring our point that models are often just fitting to dataset artifacts. We will stress that if we want to build better, more reliable evaluations, they cannot be based on such ambiguous examples.


2. On the Omission of Retrieval Tasks (Response to FCbW, RywH, zWFZ, N1ED)

All four reviewers noted the exclusion of retrieval as a major limitation, given it is a primary application of embeddings.
We agree that retrieval is critical. However, a direct human evaluation of large-scale retrieval presents unique methodological challenges. Building a robust human evaluation framework for large-scale retrieval requires particular considerations that would constitute significant work on its own; a human would need to recall or read thousands of candidate documents per query.
To address this, our study includes Reranking tasks, which we posit as the necessary and human-evaluable proxy for retrieval. We ask humans to evaluate the relevance of the top-10 candidate documents for a query, which is directly human-evaluable and conceptually equivalent in embedding-space behavior. This design follows prior work (e.g., MTEB Arena human evaluation) and ensures comparability with model reranking tasks. We will explicitly state this reasoning in Section 3.2, emphasizing that our reranking evaluation serves as the human-accessible proxy for retrieval.

3. On Methodological Robustness (Small Samples & Annotators) (Response to FCbW, RywH, zWFZ, N1ED)

All reviewers rightly raised concerns about the small sample sizes (20-50 examples) and limited annotator pool (1-2 per task), which impact statistical robustness.
We acknowledge this limitation. This study was intentionally designed as a breadth-first framework. Our primary goal was to establish the first human baselines across a wide variety of 16 MTEB tasks. Crucially, our pool is diverse in terms of linguistic and cultural background, covering five distinct languages (English, Arabic, Russian, Norwegian Bokmål, and Danish). We argue this diversity is particularly important for a multilingual benchmark, which was necessary to uncover the cross-task patterns (like the IAA-vs-superhuman-performance link) that form our central contribution.
We agree that more annotations would be ideal. However, our current "broad-and-shallow" design (with over 1,500 individual annotations) is what enables us to make our central claim about dataset quality.
To show the uncertainty range and address concerns about statistical robustness, we will add 95% confidence intervals and statistical significance testing to our main result tables. Inter-annotator agreement metrics (Fleiss’ κ, correlation-based measures for STS, etc.) are already reported in Appendix D.
Finally, we are preparing a public call for additional annotators to expand linguistic and demographic diversity in the next HUME leaderboard release, which will strengthen statistical power, generalizability and diversity.

---

### Author Response · Authors · 2025-12-03

We thank all reviewers and the Area Chair for their valuable feedback throughout the review process. We have uploaded a final revised version of our paper. This revision incorporates the constructive feedback we received during the discussion period. Key updates include:

- Statistical Robustness (All Reviewers): Added 95% confidence intervals and significance testing to all results tables, enhancing the reliability of our findings despite the breadth-first design.


- LLM-as-Annotator Evaluation (RywH, zWFZ): Conducted new experiments with 9 state-of-the-art LLMs, demonstrating that while LLMs offer scalability, they fall short of human performance, particularly on high-agreement reranking tasks (Section 4.4, Tables 10-12).


- Clarified Motivation (FCbW, N1ED): Extensively revised our framing to emphasize that human evaluation serves as a diagnostic tool for benchmark calibration, not a performance ceiling. Models achieving "superhuman" performance on low-agreement tasks likely overfit to annotation artifacts rather than demonstrate genuine semantic superiority.


- Retrieval Justification (All Reviewers): Explicitly stated in Section 3.2 that reranking serves as our human-evaluable proxy for retrieval, addressing methodological feasibility concerns.


- Ethics Section (RywH): Added Section 6.1 clarifying that annotators were consenting co-authors with diverse linguistic and cultural backgrounds across five languages.


- Practical Impact: Highlighted ongoing coordination with MTEB maintainers to operationalize findings, demonstrating concrete pathways for benchmark improvement.


We believe these additions significantly strengthen the empirical rigor of our study and address the major methodological concerns raised. We thank you again for the feedback that drove these improvements.

---

### Meta-Review · Area_Chair_Xu8A · 2026-01-07

**Summary:**

The paper asks how good humans are at MTEB (text embedding) tasks and attempts to answer it by evaluating the MTEB performance of several co-authors. They find that the average (across tasks) human performance is close to but a bit lower than the SOTA model performance. They identify tasks where human agreement is particularly low while the model performance is particularly high, suggesting possible issues with some of the MTEB tasks.

The reviewers were not convinced by the motivation, had concerns about very small sample sizes (1-2 human subjects) and lack of statistical testing and comparisons, and suggested some additional experiments. Still, several reviewers found the paper interesting and timely.

**Reviewer Concerns:**

I read the updated paper and really liked it. In my opinion, the authors did a good job addressing reviewers' concerns. In particular, I found the motivation very clearly explained and compelling, and some statistical testing and confidence intervals has been added (even though IMHO not enough, see below). The authors also conducted an entire separate experiment (Table 2) suggested by the reviewers.

Of course the extremely small sample size remains a very serious concern and could not be addressed during the rebuttal stage. The paper would be much stronger if it had ~10 human subjects. But despite that, I personally think the paper is interesting as it is.

Regarding statistical testing and CIs: I think it can be added more prominently and more coherently. The authors now compute CIs only for human performance, but the models were evaluated on the same tasks and texts, so the same CIs can be computed for each model. I would like to see error bars for all values in Figure 1, and it is unclear to me if the human performance (77.6) is actually statistically significantly below the best model performance (80.1). Similarly, in Table 1 everything that is not stat. signif. different from the best value in a column can be bold.

The experiments added in review (Table 2) are presented as something very different from the "main" set of experiments, but to me they are almost the same, so that was a bit confusing. Essentially here the authors are measuring the performance of generative LLMs at the same tasks as assessed in Figure 1. They find that generative LLMs perform worse than specialized embedding models. This could be presented more coherently. I did not understand why clustering tasks had to be excluded here... If humans could be instructed to do these tasks, then certainly ChatGPT could also be instructed to do them?

Finally, the statement that opens the Discussion ("models achieve their highest performance... precisely where human experts show the least agreement") is presented as the main finding but is not DIRECTLY visible in the results: there is no table and no figure that shows or quantifies that. A figure could show human agreement vs. model-human performance gap, or something like that. If the authors think this is the main finding, it should be presented and quantified in the Results.

**Reviewer Scores:**

The reviewers' scores were 4/4/4/4. All reviewers responded to the authors' rebuttals and none of them seemed to have increased their scores. However, they did not see the updated manuscript. Personally I feel the updated manuscript does address some of the important concerns -- even though not all. I think some of the reviewers would increase the score, but the paper would most likely still be very much on the verge. I would say the final scores could be 4/4/6/6.

As I personally liked the paper, I am going to go out on a limb and recommend acceptance. If the paper does get accepted, I strongly encourage the authors to put in more work to address the points I raised above.

---

### Decision · Program_Chairs · 2026-01-26

Accept (Poster)